# TOWARDS EQUIVARIANT GRAPH CONTRASTIVE LEARNING VIA CROSS-GRAPH AUGMENTATION

## ABSTRACT

Leading graph contrastive learning (GCL) frameworks conform to the invariance mechanism by encouraging insensitivity to different augmented views of the same graph. Despite the promising performance, invariance worsens representation when augmentations cause aggressive semantics shifts. For example, dropping the super-node can dramatically change a social network's topology. In this case, encouraging invariance to the original graph can bring together dissimilar patterns and hurt the task of instance discrimination. To resolve the problem, we get inspiration from equivariant self-supervised learning and propose Equivariant Graph Contrastive Learning (E-GCL) to encourage the sensitivity to global semantic shifts. Viewing each graph as a transformation to others, we ground the equivariance principle as a cross-graph augmentation – graph interpolation – to simulate global semantic shifts. Without using annotation, we supervise the representation of cross-graph augmented views by linearly combining the representations of their original samples. This simple but effective equivariance principle empowers E-GCL with the ability of cross-graph discrimination. It shows significant improvements over the state-of-the-art GCL models in unsupervised learning and transfer learning. Further experiments demonstrate E-GCL's generalization to various graph pre-training frameworks. Code is available at https://anonymous.4open.science/r/E-GCL/

## 1 INTRODUCTION

Graph contrastive learning (GCL) (You et al., 2020; Suresh et al., 2021; Xu et al., 2021) is a prevailing paradigm for self-supervised learning (Chen et al., 2020; Zbontar et al., 2021) on graph-structured data. It typically pre-trains a graph neural network (GNN) (Dwivedi et al., 2020) without labeled data, in an effort to learn generalizable representations and boost the fine-tuning on downstream tasks. The common theme across recent GCL studies is instance discrimination (Dosovitskiy et al., 2014; Purushwalkam & Gupta, 2020) — viewing each graph as a class of its own, and differing it from other graphs. It galvanizes representation learning to capture discriminative characteristics of graphs.

Towards this end, leading GCL works usually employ two key modules: graph augmentation and contrastive learning. Specifically, graph augmentation adopts the "intra-graph" strategy to create multiple augmented views of each graph, such as randomly dropping nodes (You et al., 2020) or adversarially perturbing edges (Suresh et al., 2021). The views stemming from the same graph constitute the positive samples of this class, while the views of other graphs are treated as negatives. Consequently, contrastive learning encourages the agreement between positive samples and the discrepancy between negatives. This procedure essentially imposes "invariance" (Purushwalkam & Gupta, 2020; Dangovski et al., 2022) upon representations — making the anchor graph's representation invariant to its intra-graph augmentations (Figure 1a). Formally, let $g$ be the anchor graph, $\mathcal{P}$ be the groups of intra-graph augmentations, and $\phi(\cdot)$ be the GNN encoder. The "invariance to intra-graph augmentations" mechanism states $\phi(g) = \phi(T_p(g)), \forall p \in \mathcal{P}$ — the representation $\phi(g)$ is insensitive to the changes in augmentation $p$, where $T_p(g)$ is the action of augmentation $p$ on graph $g$. We refer to works adopting this mechanism as Invariant Graph Contrastive Learning (I-GCL).

However, we argue that invariance to intra-graph augmentations alone is insufficient to improve the semantic quality of graph representations and boost the downstream performance:

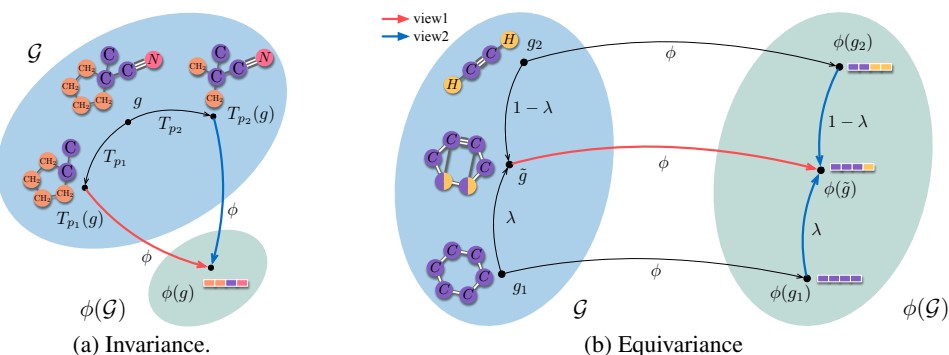

(a) Invariance.  (b) Equivariance

Figure 1: (a) invariance to intra-graph augmentations; (b) equivariance to cross-graph augmentations.

- Limiting the augmentations to local substructures of an individual graph is aggressive (Purushwalkam & Gupta, 2020; Wang et al., 2022) insofar as the augmented views fragmentarily or even wrongly describe the characteristics of the anchor graph. Take a molecule graph as an example. After randomly dropping some nodes, one view could hold a cyano group ($-C{\equiv}N$) that determines the property of molecule hypertoxic, while another could corrupt this functional group. Thus, intra-graph augmentations are inadequate for presenting a holistic view of the anchor graph.

- Worse still, aggressive augmentations easily make two positive views far from each other, but the invariance mechanism blindly forces their representations to be invariant. Considering the molecule graph's views again (*cf.* Figure 1a), invariance-guided contrastive learning simply maximizes their representation agreement, regardless of the changes in the hypertoxic property. Therefore, it might amplify the negative impact of aggressive intra-graph augmentations and restrain representations from reflecting the instance semantics faithfully.

To mitigate these negative influences, we get inspiration from the recent work on equivariant self-supervised learning (E-SSL) (Dangovski et al., 2022). It splits the augmentations into two parts, to which representations should be insensitive and sensitive, and then establishes the invariance and equivariance mechanisms correspondingly. The idea of "equivariance" is our focus, which makes representations aware of semantic changes caused by certain augmentations $\mathcal{H}$. Here we formulate it as $\phi(T_h(g)) = T'_h(\phi(g)), \forall h \in \mathcal{H}$, where $T_h(g)$ and $T'_h(\phi(g))$ are the actions of augmentation $h$ on graph $g$ and representation $\phi(g)$, respectively. Jointly learning equivariance to sensitive augmentations $\mathcal{H}$ and invariance to insensitive augmentations $\mathcal{P}$ is promising to shield representations from the harms of aggressive augmentations. Nonetheless, it is hard, without domain knowledge (Dangovski et al., 2022; Chuang et al., 2022) or extensive testing (Dangovski et al., 2022), to tell apart sensitive and insensitive augmentations.

To embody equivariance in GCL, we propose a simple but effective approach of Equivariant Graph Contrastive Learning (E-GCL). E-GCL is an instantiation of E-SSL for graphs. Unlike previous E-SSL works, E-GCL leaves existing intra-graph augmentations untouched, and creates new augmentations through the "cross-graph" strategy. Concretely, inspired by mixup (Guo & Mao, 2021; Zhang et al., 2018), the cross-graph augmentation interpolates the raw features of two graphs (*i.e.,* $T_h$), while employing the same interpolation strategy on the graph labels that are portrayed by graph representations (*i.e.,* $T'_h$). The augmentations across graphs not only maintain the holistic information on self-discrimination, but also are orthogonal to the intra-graph augmentations. On the top of intra- and cross-graph of augmentations, E-GCL separately builds the invariance and equivariance principles to guide the representation learning. The equivariance to cross-graph augmentations diminishes the harmful invariance to aggressive augmentations that change global semantics. Integrating two principles enables representations to be sensitive to global semantic shifts across different graphs and insensitive to local substructure perturbations of single graphs. Experiments show that E-GCL achieves promising performances to surpass current state-of-the-art GCL models, across diverse settings. We also demonstrate E-GCL's generalization to various SSL frameworks, including BarlowTwins (Zbontar et al., 2021), GraphCL (You et al., 2020) and SimSiam (Chen & He, 2021).

## 2 PRELIMINARIES: INVARIANT GRAPH CONTRASTIVE LEARNING

We begin by presenting the instance discrimination task and the invariance mechanism of I-GCL, and then introduce two key ingredients: graph augmentations and contrastive learning.

**Instance Discrimination.** Let $\mathcal{G} = \{g_n\}_{n=1}^N$ be the set of unlabeled graph instances. We denote a graph instance $g \in \mathcal{G}$ by $(\mathcal{V}, \mathcal{E})$ involving the node set $\mathcal{V}$ and the edge set $\mathcal{E}$. This graph structure can be represented as an adjacency matrix $\mathbf{A} \in \{0,1\}^{|\mathcal{V}| \times |\mathcal{V}|}$, where $A_{uv} = 1$ if the edge $(u, v) \in \mathcal{E}$ from node $u$ to node $v$ holds, otherwise $A_{uv} = 0$. Moreover, each node $v \in \mathcal{V}$ could have $d_1$-dimensional features $\mathbf{x}_v \in \mathbb{R}^{d_1}$, while each edge $(u, v) \in \mathcal{E}$ might have $d_2$-dimensional features $\mathbf{e}_{uv} \in \mathbb{R}^{d_2}$.

On the graph data $\mathcal{G}$ without annotations, contrastive self-supervised learning (SSL) aims to pre-train a graph encoder $\phi : \mathcal{G} \to \mathbb{R}^d$ that projects the graph instances to a $d$-dimensional space, so as to enhance the encoder's representation ability and facilitate its fine-tuning in downstream tasks. Towards this end, a prevailing task of pre-training is instance discrimination (Dosovitskiy et al., 2014; Purushwalkam & Gupta, 2020; Li et al., 2021) — treating each graph instance as one single class, and distinguishing it from the other graph instances.

**Invariance.** A leading solution to instance discrimination is to maximize the representation agreement between augmented views of the same graph, while minimizing the representation agreement between views of two different graphs. It essentially encourages each instance's representation to be invariant to the augmentations (Dangovski et al., 2022; Grill et al., 2020; Zbontar et al., 2021). Mathematically, invariance can be described by groups (Dangovski et al., 2022; Kondor & Trivedi, 2018; Maron et al., 2019a). Let $\mathcal{P}$ be a group of augmentations (*aka.* transformations). Invariance makes the encoder $\phi$ insensitive to the actions $T : \mathcal{P} \times \mathcal{G} \to \mathcal{G}$ of the group $\mathcal{P}$ on the graphs $\mathcal{G}$, formally:

$$\phi(g) = \phi(T_p(g)), \quad \forall p \in \mathcal{P}, \ \forall g \in \mathcal{G}, \tag{1}$$

where $T_p(g) := T(p, g)$ is an action of applying the augmentation $p$ on the instance $g$. Dictating invariance to the encoder will output the same representations for the original and augmented graphs. Probing into Equation (1), we find two key ingredients: intra-graph augmentation and contrastive learning, and will present their common practices in prior studies.

**Intra-graph Augmentation.** Typically, the augmentation group $\mathcal{P}$ is pre-determined to imply prior knowledge of graph data. Early studies (Hu et al., 2020; Qiu et al., 2020; You et al., 2020; Zhu et al., 2020) instantiate augmentations as randomly corrupting the topological structure, node features, or edge features of individual graph instances. For example, AttrMasking (Hu et al., 2020) masks node and edge attributes, and applies an objective to reconstruct them. GCC (Qiu et al., 2020) explores random walks over the anchor graph to create different subgraph views. GraphCL (You et al., 2020) systematically investigates the combined effect of various random augmentations. Despite the success, random corruptions are too aggressive to maintain the semantic consistency (Guo & Mao, 2021) between the anchor graph and its augmented views. The invariance principle blindly ignores the semantic shift, thus easily pushing dissimilar patterns together and making a pernicious impact on the representation learning. Some follow-on studies (Zhu et al., 2021; Subramonian, 2021; Suresh et al., 2021; Xu et al., 2021) learn augmentations instead to underscore salient substructures, so as to mitigate the semantic shift. For instance, GCA (Zhu et al., 2021) applies node centralities to discover important substructures in social networks. MICRO-Graph (Subramonian, 2021) learns chemically meaningful motifs to help the informative subgraph sampling. More recently, AD-GCL (Suresh et al., 2021) adopts the idea of information bottleneck to adversarially learn the salient subgraphs.

**Contrastive Learning.** Upon the augmented views, the contrastive learning objective is to classify whether they come from identical instances. Specifically, it pulls the augmented views derived from the same instance (*i.e.,* positive samples) together and pushes the views of different instances (*i.e.,* negative samples) apart (Chen et al., 2020; He et al., 2020). The common practices of this objective are InfoNCE (van den Oord et al., 2018), NCE (Misra & van der Maaten, 2020), and NT-Xent (Chen et al., 2020). Here we consider the NT-Xent adopted by GraphCL. Given a minibatch of graph instances $\{g_i\}_{i=1}^N$, it first generates two different augmented views, denoted as $\{g_i^1 | g_i^1 = T_{p_1}(g), p_1 \sim \mathcal{P}\}_{i=1}^N$ and $\{g_i^2 | g_i^2 = T_{p_2}(g), p_2 \sim \mathcal{P}\}_{i=1}^N$, and then feeds them into the encoder to yield the representations as $\{\mathbf{z}_i^1 | \mathbf{z}_i^1 = \rho(\phi(g_i^1))\}_{i=1}^N$ and $\{\mathbf{z}_i^2 | \mathbf{z}_i^2 = \rho(\phi(g_i^2))\}_{i=1}^N$, where $\rho(\cdot)$ is an MLP projection head. Formally, the loss of NT-Xent is:

$$\ell(\{\mathbf{z}_i^1\}_{i=1}^N, \{\mathbf{z}_i^2\}_{i=1}^N) = -\frac{1}{N} \sum_{i=1}^N \log \frac{\exp(s(\mathbf{z}_i^1, \mathbf{z}_i^2)/\tau)}{\sum_{j=1, j \neq i}^N \exp(s(\mathbf{z}_i^1, \mathbf{z}_j^2)/\tau)}, \tag{2}$$

where $s(\cdot)$ is the function of cosine similarity, and $\tau$ is a temperature hyperparameter.

In a nutshell, the interplay between intra-graph augmentation and contrastive learning is tailor-made for invariance to make the encoder insensitive to differences between the anchor and augmented

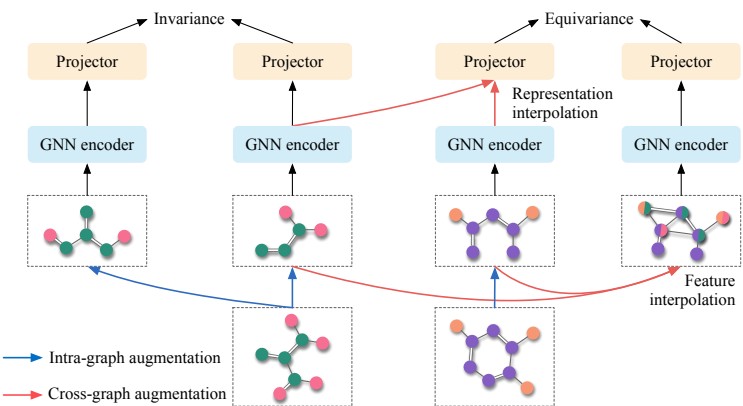

Figure 2: The framework of E-GCL. The GNN encoder learns invariance for intra-graph augmentation of dropNode and equivariance for cross-graph augmentation of graph interpolation.

views. In this work, we explore equivariance on cross-graph augmentations to make the encoder sensitive to the changes in self-discriminative information.

## 3 METHODOLOGY: EQUIVARIANT GRAPH CONTRASTIVE LEARNING

Here we present the E-GCL framework, which imposes two principles — invariance to intra-graph augmentations (Figure 2 left) and equivariance to cross-graph augmentations (Figure 2 right) — on the representation learning, aiming to mitigate the potential limitations of I-GCL. Next, we start with the concepts of equivariance and cross-graph augmentations.

### 3.1 EQUIVARIANCE

Inspired by the recent E-SSL studies (Dangovski et al., 2022; Chuang et al., 2022), we aim to patch invariance's potential limitations with equivariance. Mathematically, with a group of augmentations $\mathcal{H}$, the encoder $\phi(\cdot)$ is said to be $\mathcal{H}$-equivariant *w.r.t.* the actions $T : \mathcal{H} \times \mathcal{G} \to \mathcal{G}$ and $T' : \mathcal{H} \times \mathbb{R}^d \to \mathbb{R}^d$ of the group $\mathcal{H}$ applied on the graph space $\mathcal{G}$ and the representation space $\mathbb{R}^d$, if

$$\phi(T_h(g)) = T'_h(\phi(g)), \quad \forall h \in \mathcal{H}, \ \forall g \in \mathcal{G}, \tag{3}$$

where $T_h(g)$ is the action of applying the transformation $h$ on the graph instance $g$, while $T'_h(\phi(g))$ is the action of $h$ on the representation $\phi(g)$. $\mathcal{H}$-equivariant requires that, given a transformation $h \in \mathcal{H}$, $h$'s influence on the graph should be faithfully reflected by the change of the graph's representation. Taking Figure 1b as an example, given the graph's global semantics are perturbed by graph interpolation, the representation yielded by the equivariant encoder should transform in a definite way. Jointly analyzing Equations (1) and (3), it can be shown that invariance is a special case of equivariance when setting $T'_h$ as the identity mapping. However, generalizing GCL to equivariance remains unexplored, thus a focus of our work.

Furthermore, as suggested in the recent E-SSL studies (Dangovski et al., 2022; Chuang et al., 2022), jointly imposing invariance to some transformations $\mathcal{P}$ and equivariance to other transformations $\mathcal{H}$ is promising to result in better representations than relying solely on one of them. Here we term $\mathcal{P}$ and $\mathcal{H}$ as insensitive and sensitive transformations, respectively. For example, in computer vision, E-SSL (Dangovski et al., 2022) sets grayscale of images as $\mathcal{P}$, while treating rotations as $\mathcal{H}$; in natural language processing, DiffCSE (Chuang et al., 2022) treats the model dropout as $\mathcal{P}$, while using the word replacement as $\mathcal{H}$. Clearly, it is of crucial importance to partition augmentations into $\mathcal{P}$ and $\mathcal{H}$. Nonetheless, these studies either conduct extensive testings on the impact of different partitions (Dangovski et al., 2022) which is time-consuming, or exploit domain knowledge to heuristically partition (Chuang et al., 2022) which might generalize poorly to other domains. Hence, it is infeasible to apply these strategies on graph augmentations. Worse still, different graph augmentations stem mostly from the perturbation of graph structures, thus highly likely to corrupt the same attributes of graphs. Taking the graph $g$ in Figure 1a as an example, masking the nitrogen $N$ atom or dropping the $C \equiv N$ bond will both corrupt the cyano group and break the corresponding molecular properties. In a nutshell, owing to (1) the common paradigm of structure corruption and (2) the risk of categorizing

them all as insensitive augmentations, we conservatively argue that it is hard to partition graph augmentations into sensitive $\mathcal{H}$ and insensitive parts $\mathcal{P}$.

In this work, leaving partitioning untouched, we remain intra-graph augmentations as the insensitive transformations $\mathcal{P}$ and propose new augmentations across graphs as the sensitive transformations $\mathcal{H}$.

## 3.2 CROSS-GRAPH AUGMENTATION

We first introduce graph interpolation (Guo & Mao, 2021) to create cross-graph augmentations as $\mathcal{H}$. Different from previous work, we propose an extension of graph interpolation for SSL (Section 3.3). We also connect it to group theory and address its limitation of sensitivity to the relative permutation.

**Interpolating Graphs as Cross-graph Augmentations.** Given two graph instances $g \in \mathcal{G}$ and $g' \in \mathcal{G}$, we employ mixup (Zhang et al., 2018), a simple yet effective linear interpolation approach, on the input features and class labels, respectively:

$$\tilde{g} = \lambda g + (1 - \lambda)g', \quad \tilde{y} = \lambda y + (1 - \lambda)y', \tag{4}$$

where $y$ and $y'$ separately denote the one-hot encodings to indicate the instance identities of $g$ and $g'$ in the instance discrimination task; $\lambda \sim \text{Beta}(\alpha, \alpha) \in [0, 1]$ is the interpolation ratio sampled from a Beta distribution, in which $\alpha$ is a hyperparameter. This mixup strategy is initially proposed for supervised learning, aiming to put the interpolated samples in-between different classes and make the decision boundary robust to slightly corrupted samples (Verma et al., 2019; Zhang et al., 2018; 2021). Despite the success of mixing image and text, it is challenging to interpolate graphs due to the structural differences between graph instances (*e.g.,* varying topologies and sizes).

To this end, we draw inspiration from the recent work (Guo & Mao, 2021) to perform linear interpolation between graphs. Specifically, with $g = (\mathcal{V}, \mathcal{E})$ and $g' = (\mathcal{V}', \mathcal{E}')$, we mitigate their structural differences by padding virtual nodes and edges, which are associated with zero features $\mathbf{0}$. Assuming $|\mathcal{V}| \leq |\mathcal{V}'|$, $g$ can be updated as a new graph with $|\mathcal{V}'|$ nodes, where the original node set $\mathcal{V}$ remains unchanged but adds $|\mathcal{V}'| - |\mathcal{V}|$ dummy virtual nodes, and the original nodes connect the virtual nodes with dummy virtual edges. Having padded two graphs to the same size, now we can directly add them up. Before the interpolation, we first merge two node and edge sets as the new ones: $\tilde{\mathcal{V}} = \mathcal{V} \cup \mathcal{V}'$, $\tilde{\mathcal{E}} = \mathcal{E} \cup \mathcal{E}'$. Then, $\tilde{g} = \lambda g + (1 - \lambda)g'$ in Equation (4) is achieved by exerting linear interpolation on the adjacency matrices, node features, and edge features:

$$\tilde{\mathbf{A}} = \lambda \mathbf{A} + (1 - \lambda)\mathbf{A}', \quad \tilde{\mathbf{x}}_v = \lambda \mathbf{x}_v + (1 - \lambda)\mathbf{x}'_v, \quad \tilde{\mathbf{e}}_{uv} = \lambda \mathbf{e}_{uv} + (1 - \lambda)\mathbf{e}'_{uv}, \tag{5}$$

where $\mathbf{A}$ and $\mathbf{A}'$ are the adjacency matrices of $g$ and $g'$ after padding; $\mathbf{x}_v$ and $\mathbf{x}'_v$ are the features of node $v \in \tilde{\mathcal{V}}$, which separately come from $g$ and $g'$; similarly, $\mathbf{e}_{uv}$ and $\mathbf{e}'_{uv}$ separately denote the features of edge $(u, v) \in \tilde{\mathcal{E}}$ from $g$ and $g'$. Consequently, we generate a cross-graph augmentation.

**Connecting Cross-graph Augmentations to Groups.** In the language of groups, we can describe the cross-graph augmentation in Equation (4) as a group of transformations. Given the input $(\lambda, g, g')$, we can systemize the graph interpolation as two steps: (1) feature rescaling: $\hat{g} = \lambda g$, which rescales the node and edge features of $g$ with the ratio $\lambda$; (2) instance composition: $\tilde{g} = C(\hat{g}, \hat{g}') = \hat{g} + \hat{g}'$, which adds the other rescaled graph $\hat{g}' = (1 - \lambda)g'$. To construct a closed space for graph interpolation, we first define $\hat{\mathcal{G}} = \{\hat{g} | \lambda \in [-1, 1], g \in \mathcal{G}\}$ by performing feature rescaling on $\mathcal{G}$ to enable direct sampling of rescaled graphs. We allow $\lambda < 0$ to include the inverse elements of graphs. Then, we generate $\mathcal{I} = < \hat{\mathcal{G}} >$ by combining the graphs in $\hat{\mathcal{G}}$ via instance composition. We show that $(\mathcal{I}, C)$ forms a group in Appendix A.1. It is worth noting that each instance can be viewed as a transformation to others, *i.e.,* $C(g, \cdot) := C_g(\cdot)$, such that semantic shifts can be described via algebraic operators.

**Group Averaging for Insensitivity to Relative Permutation.** Note that Equation (5) can output different interpolations, when the node orders of one graph or padding positions of dummy nodes change. We ascribe operations on "node orders" and "padding positions" to the "relative permutation" between two input graphs. Unlike images and texts, the canonical permutations (orderings) of nodes in graphs are unlabeled. The default node permutations encode nothing useful about graph semantics (Xu et al., 2019; Hamilton et al., 2017). Thus, we enforce insensitivity to relative permutations by randomly permuting nodes in the bigger graph before graph interpolation. Next, we justify and develop this design with group averaging.

Group averaging can make known architectures invariant to new symmetries (Puny et al., 2022; Yarotsky, 2022). It can be used for strict invariance to relative permutations. Let $P \sim S_n$ be a random

permutation, and $T_P \circ g$ be permuting graph $g$ by $P$. Assuming $|g| \geq |g'|$, we obtain the graph interpolation as $\lambda T_P \circ g + (1 - \lambda)g'$ and its representation as $\phi(\lambda T_P \circ g + (1 - \lambda)g')$. To achieve strict invariance to relative permutations, we apply group averaging on the permutation operators:

$$\Phi(\lambda, g, g') = \frac{1}{|S_n|} \sum_{P \in S_n} \phi(\lambda T_{P^{-1}} \circ g + (1 - \lambda)g'), \tag{6}$$

where $\Phi$ is the function of group averaging. $\Phi$ is invariant to relative permutations between $g$ and $g'$, in the sense that $\Phi(\lambda, g, g') = \Phi(\lambda, T_P \circ g, g') = \Phi(\lambda, g, T_{P'} \circ g')$ for all $P, P' \sim S_n$. Intuitively, it is achieved by averaging over all relative permutations. See Appendix A.2 for the proof. However, the intractability of averaging over $S_n$ naturally arises as a problem. Following Murphy et al. (2019), our random permutation strategy $\phi(\lambda T_P \circ g + (1 - \lambda)g')$ is an unbiased estimator of $\Phi$. Further, this strategy optimizes $\rho \circ \phi$ toward an optima insensitive to the relative permutation. By Proposition 1, we conclude that using random permutation is a tractable surrogate for optimizing an invariant network to relative permutation. Appendix D.2 shows the influence of sampled permutation numbers. For simplicity, we still use $\lambda g + (1 - \lambda)g'$ to represent graph interpolation in the rest text.

**Proposition 1.** *The contrastive loss with $\phi(\lambda T_P \circ g + (1 - \lambda)g')$ upper bounds the loss of an invariant network to relative permutation $\frac{1}{|S_n|} \sum_{P \in S_n} \rho(\phi(\lambda_i T_P \circ g_i + (1 - \lambda_i)g'_i))$.*

### 3.3 EQUIVARIANCE TO CROSS-GRAPH AUGMENTATIONS

Revisiting Equation (4), we can find the two terms of interpolation strategy align with the equivariance mechanism in Equation (3). The semantic change caused by the feature interpolation is equivalently reflected by the label interpolation. Hence, we can instantiate the equivariance mechanism based on the feature and label interpolations. In the SSL setting, we interpolate graph representations as the alternative for label interpolation. Graph representations are derived from the encoder $\phi(\cdot)$ with a global readout layer to summarize the graphs' global semantics. Hence, as shown by the right side of Figure 2, we can parameterize equivariance approximately as:

$$\phi(\lambda g + (1 - \lambda)g') \approx \lambda \phi(g) + (1 - \lambda)\phi(g'). \tag{7}$$

Minimizing the distance between Equation (7)'s two sides allows the encoder to improve sensitivity to the global semantic shifts caused by cross-graph augmentations.

Although the strict equivariance is hardly guaranteed, experiments show that approaching Equation (7) can boost the performance on downstream tasks (*cf.* Section 4.2). Furthermore, if the encoder is powerful enough to distinguish the interpolated graphs, it has a deterministic reflection in the representation space $\phi(\mathcal{G})$ for a fixed transformation $C(g, \cdot)$ in graph space $\mathcal{G}$ (Non-trivial Equivariance (Dangovski et al., 2022)). See Appendix A.1 for the proof.

**Proposition 2.** *Assuming the encoder can detect the isomorphism of interpolated graphs, there exists a GNN encoder $\phi$ that is non-trivially equivariant to the graph interpolation transformation.*

### 3.4 IMPLEMENTING E-GCL

This section details our implementation of E-GCL (Figure 2). Specifically, given a minibatch of graph instances $\{g_i\}_{i=1}^N$, we impose (1) invariance to intra-graph augmentation and (2) equivariance to cross-graph augmentation simultaneously on the shared encoder.

**Invariance.** For the invariance principle, we follow the I-GCL paradigm to resort to the standard intra-graph augmentations $\mathcal{P}$ (*e.g.,* randomly dropping nodes in GraphCL), and create two augmented views of individual graphs: $\{g_i^1 | g_i^1 = T_{p_1}(g), p_1 \sim \mathcal{P}\}_{i=1}^N$ and $\{g_i^2 | g_i^2 = T_{p_2}(g), p_2 \sim \mathcal{P}\}_{i=1}^N$. Consequently, the encoder $\phi(\cdot)$ brings forth two representation lists: $\{\mathbf{z}_i^1 | \mathbf{z}_i^1 = \rho(\phi(g_i^1))\}_{i=1}^N$ and $\{\mathbf{z}_i^2 | \mathbf{z}_i^2 = \rho(\phi(g_i^2))\}_{i=1}^N$, in which $\rho(\cdot)$ is an MLP projector.

**Equivariance.** For the equivariance principle, we first randomly shuffle the graphs in $\{g_i^2\}_{i=1}^N$, termed as $\{g_{\pi(i)}^2\}_{i=1}^N$, where $\pi : [N] \to [N]$ is the function of random shuffling. Following the left side of Equation (7) and applying random permutations $P_i \sim S_n$ for all $i \in [N]$, we create the feature interpolations and then generate their representations as $\{\mathbf{z}_i^3 | \mathbf{z}_i^3 = \rho(\phi(\lambda T_{P_i} \circ g_i^1 + (1 - \lambda)\lambda g_{\pi(i)}^2))\}_{i=1}^N$. Meanwhile, according to the right side of Equation (7), we arrive at the representation interpolations as $\{\mathbf{z}_i^4 | \mathbf{z}_i^4 = \rho(\lambda \phi(g_i^1) + (1 - \lambda)\phi(\lambda g_{\pi(i)}^2))\}_{i=1}^N$.

Table 1: Main experiment performances. ∗ denotes our reproduced results using the released codes. Other baseline results are borrowed from the original papers. **Bold** indicates the best performance and underline indicates the second best performance.

(a) Unsupervised learning accuracies (%) on the TU datasets.

| Dataset | NCI1 | PROTEINS | DD | MUTAG | COLLAB | RDT-B | RDT-M5K | IMDB-B | AVG | GAIN |
|---|---|---|---|---|---|---|---|---|---|---|
| No Pre-train* | $72.67_{\pm1.16}$ | $73.81_{\pm0.30}$ | $77.01_{\pm0.90}$ | $84.26_{\pm1.16}$ | $62.92_{\pm0.02}$ | $72.45_{\pm0.47}$ | $45.32_{\pm0.30}$ | $67.54_{\pm0.27}$ | 69.50 | - |
| InfoGraph* | $78.27_{\pm0.64}$ | $74.56_{\pm0.57}$ | $77.01_{\pm0.90}$ | $87.87_{\pm1.85}$ | $70.73_{\pm0.48}$ | $83.22_{\pm2.78}$ | $55.82_{\pm0.29}$ | $70.96_{\pm0.60}$ | 74.81 | 5.31 |
| GraphCL* | $79.25_{\pm0.40}$ | $74.50_{\pm0.85}$ | $78.24_{\pm0.99}$ | $87.65_{\pm1.66}$ | $71.59_{\pm0.52}$ | $90.54_{\pm0.30}$ | $55.67_{\pm0.45}$ | $71.38_{\pm0.40}$ | 76.10 | 6.60 |
| JOAO* | $78.55_{\pm0.17}$ | $74.39_{\pm0.80}$ | $77.44_{\pm0.85}$ | $88.85_{\pm1.51}$ | $70.52_{\pm0.63}$ | $88.49_{\pm0.76}$ | $56.04_{\pm0.24}$ | $71.44_{\pm0.52}$ | 75.72 | 6.22 |
| AD-GCL* | $72.95_{\pm0.45}$ | $73.62_{\pm0.63}$ | $76.06_{\pm0.44}$ | $\mathbf{89.25_{\pm1.29}}$ | $70.70_{\pm0.46}$ | $87.03_{\pm1.18}$ | $54.81_{\pm0.42}$ | $71.62_{\pm0.66}$ | 74.51 | 5.01 |
| GraphMAE* | $75.00_{\pm0.95}$ | $73.92_{\pm0.97}$ | $76.15_{\pm0.99}$ | $87.17_{\pm1.02}$ | $72.92_{\pm3.88}$ | $81.27_{\pm2.51}$ | $49.63_{\pm1.67}$ | $71.96_{\pm0.65}$ | 73.50 | 4.00 |
| R-GCL* | $78.79_{\pm0.42}$ | $\underline{74.60_{\pm0.71}}$ | $\mathbf{79.14_{\pm0.39}}$ | $88.12_{\pm1.38}$ | $71.44_{\pm0.60}$ | $90.11_{\pm0.41}$ | $55.96_{\pm0.42}$ | $\underline{71.84_{\pm0.76}}$ | $\underline{76.25}$ | 6.75 |
| E-GCL | $\mathbf{79.95_{\pm0.25}}$ | $\mathbf{75.18_{\pm0.40}}$ | $77.83_{\pm0.52}$ | $88.20_{\pm0.85}$ | $\mathbf{74.63_{\pm0.28}}$ | $\mathbf{90.71_{\pm0.57}}$ | $\mathbf{56.90_{\pm0.29}}$ | $\mathbf{72.02_{\pm0.90}}$ | $\mathbf{76.93}$ | 7.43 |

(b) Transfer learning ROC-AUC (%) scores on the MoleculeNet. GTS denotes GraphTrans.

| Dataset | BBBP | Tox21 | ToxCast | SIDER | ClinTox | MUV | HIV | BACE | AVG | GAIN |
|---|---|---|---|---|---|---|---|---|---|---|
| No Pre-train* | $67.8_{\pm1.6}$ | $73.9_{\pm0.9}$ | $62.4_{\pm0.4}$ | $58.3_{\pm1.5}$ | $62.6_{\pm4.4}$ | $73.4_{\pm2.7}$ | $76.5_{\pm1.7}$ | $76.8_{\pm2.8}$ | 69.0 | - |
| Infomax* | $68.5_{\pm1.1}$ | $75.4_{\pm0.3}$ | $62.6_{\pm0.3}$ | $58.6_{\pm0.7}$ | $71.2_{\pm2.5}$ | $73.1_{\pm1.9}$ | $76.8_{\pm1.0}$ | $74.4_{\pm1.1}$ | 70.1 | 1.1 |
| ContextPred* | $72.2_{\pm1.1}$ | $75.6_{\pm0.6}$ | $63.5_{\pm0.3}$ | $60.6_{\pm0.9}$ | $70.2_{\pm2.6}$ | $74.3_{\pm1.4}$ | $77.6_{\pm0.5}$ | $79.0_{\pm0.9}$ | 71.6 | 2.6 |
| GraphCL | $69.7_{\pm0.7}$ | $73.9_{\pm0.7}$ | $62.4_{\pm0.6}$ | $60.5_{\pm0.9}$ | $76.0_{\pm2.6}$ | $69.8_{\pm2.7}$ | $\mathbf{78.5_{\pm1.2}}$ | $75.4_{\pm1.4}$ | 70.8 | 1.8 |
| JOAO | $71.4_{\pm0.9}$ | $74.3_{\pm0.6}$ | $63.2_{\pm0.5}$ | $60.5_{\pm0.7}$ | $81.0_{\pm1.6}$ | $73.7_{\pm1.0}$ | $77.5_{\pm1.2}$ | $75.5_{\pm1.3}$ | 72.1 | 3.1 |
| ADGCL* | $70.5_{\pm1.8}$ | $74.5_{\pm0.7}$ | $63.0_{\pm0.5}$ | $59.1_{\pm0.9}$ | $78.5_{\pm3.7}$ | $71.5_{\pm2.2}$ | $75.9_{\pm1.4}$ | $74.0_{\pm2.2}$ | 70.9 | 1.9 |
| GraphLOG* | $71.0_{\pm1.1}$ | $74.9_{\pm0.4}$ | $62.8_{\pm0.5}$ | $59.7_{\pm0.9}$ | $76.9_{\pm1.9}$ | $70.8_{\pm2.0}$ | $75.8_{\pm1.4}$ | $\mathbf{82.9_{\pm0.9}}$ | 71.8 | 2.8 |
| GraphMAE* | $72.2_{\pm0.9}$ | $75.1_{\pm0.4}$ | $63.0_{\pm0.3}$ | $58.5_{\pm0.7}$ | $80.5_{\pm2.0}$ | $75.7_{\pm1.2}$ | $76.4_{\pm0.8}$ | $81.3_{\pm1.0}$ | 72.8 | 3.8 |
| RGCL* | $71.2_{\pm0.9}$ | $75.3_{\pm0.5}$ | $63.1_{\pm0.3}$ | $61.2_{\pm0.6}$ | $\mathbf{85.0_{\pm0.8}}$ | $73.1_{\pm1.2}$ | $77.3_{\pm0.8}$ | $75.7_{\pm1.3}$ | 72.7 | 3.7 |
| E-GCL | $\mathbf{72.3_{\pm0.6}}$ | $74.9_{\pm0.7}$ | $64.0_{\pm0.3}$ | $\mathbf{62.8_{\pm0.5}}$ | $83.1_{\pm2.5}$ | $78.8_{\pm0.8}$ | $76.3_{\pm0.6}$ | $78.1_{\pm1.1}$ | $\underline{73.8}$ | 4.8 |
| E-GCL, GTS | $\mathbf{72.3_{\pm0.8}}$ | $\mathbf{77.9_{\pm0.6}}$ | $\mathbf{66.0_{\pm0.6}}$ | $62.4_{\pm1.0}$ | $\underline{80.7_{\pm3.0}}$ | $\mathbf{79.4_{\pm2.1}}$ | $77.8_{\pm1.1}$ | $79.7_{\pm2.4}$ | $\mathbf{74.5}$ | 5.5 |

**Cooperative Game between Invariance and Equivariance.** Based on the representations, we optimize the invariance and equivariance losses together with a weighting hyperparameter $\omega \in [0, 1]$:

$$\mathcal{L}_{\text{E-GCL}} = (1 - \omega) \cdot \underbrace{\ell(\{\mathbf{z}_i^1\}_{i=1}^N, \{\mathbf{z}_i^2\}_{i=1}^N)}_{\text{invariance loss}} + \omega \cdot \underbrace{\ell(\{\mathbf{z}_i^3\}_{i=1}^N, \{\mathbf{z}_i^4\}_{i=1}^N)}_{\text{equivariance loss}}, \tag{8}$$

where $\ell(\cdot, \cdot)$ is the loss encouraging the insensitivity between augmented views of the same graph, which is determined by the SSL backbone, such as NT-Xent (*cf.* Equation (2)) adopted by the GraphCL. Beyond contrastive learning, E-GCL is also applicable to various other SSL backbones, including BarlowTwins and SimSiam. In a nutshell, the invariance loss underscores the insensitivity to intra-graph augmentations, while the equivariance loss induces the sensitivity to cross-graph augmentations. The cooperative game between these two losses helps resolve the potential limitations of the conventional I-GCL paradigm, thus improving the expressive power of the encoder.

## 4 EXPERIMENT

In this section, we conduct experiments to answer the following research questions: **RQ1:** How effective is the proposed E-GCL in graph representation learning, and how does it generalize to existing SSL frameworks? **RQ2:** What are the properties of E-GCL and the effects of its components?

In Appendix D.1, we present more ablation studies about 1) using $\mathcal{G}$-mixup (Han et al., 2022), 2) interpolating representations at different positions, and 3) interpolating large and small graphs.

### 4.1 EXPERIMENTAL SETUP

Here we briefly introduce the baselines, datasets and evaluations. Details are in Appendix E. For a fair comparison, E-GCL uses the same intra-graph augmentation as GraphCL. If not noted, E-GCL employs a BarlowTwins backbone. We study E-GCL's generalization to other SSL frameworks later.

**Baselines**. We compare E-GCL with the following state-of-the-art graph pre-training methods: Infomax (Veličković et al., 2019), InfoGraph (Sun et al., 2020), ContextPred (Hu et al., 2020), GraphCL (You et al., 2020), JOAO (You et al., 2021), AD-GCL (Suresh et al., 2021), GraphLOG (Xu et al., 2021), GraphMAE (Hou et al., 2022), and RGCL (Li et al., 2022).

**Unsupervised Learning** evaluates the pre-trained GNNs for prediction on the same dataset. Following (You et al., 2020), we evaluate E-GCL on the eight TU datasets, including biochemical graphs

Table 2: Generalization to diverse SSL frameworks. Red denotes equivariance improves performance.

(a) Unsupervised learning accuracies (%).

| Dataset | NCI1 | PROTEINS | DD | MUTAG | COLLAB | RDT-B | RDT-M5K | IMDB-B | AVG | GAIN |
|---|---|---|---|---|---|---|---|---|---|---|
| GraphCL | $79.25_{\pm0.40}$ | $74.50_{\pm0.85}$ | $78.24_{\pm0.99}$ | $87.65_{\pm1.66}$ | $71.59_{\pm0.52}$ | $90.54_{\pm0.30}$ | $55.67_{\pm0.45}$ | $71.38_{\pm0.40}$ | 76.10 | - |
| +Equivariance | $80.22_{\pm0.38}$ | $74.57_{\pm0.46}$ | $79.15_{\pm0.98}$ | $89.79_{\pm1.07}$ | $72.75_{\pm0.66}$ | $91.28_{\pm0.32}$ | $55.80_{\pm0.24}$ | $71.70_{\pm0.49}$ | 76.91 | 0.81 |
| BarlowTwins | $79.60_{\pm0.42}$ | $74.90_{\pm0.47}$ | $77.22_{\pm0.91}$ | $86.92_{\pm1.87}$ | $72.94_{\pm0.62}$ | $90.11_{\pm0.85}$ | $55.40_{\pm0.47}$ | $71.28_{\pm0.58}$ | 76.05 | - |
| +Equivariance | $79.95_{\pm0.25}$ | $75.18_{\pm0.40}$ | $77.83_{\pm0.52}$ | $88.20_{\pm0.85}$ | $74.63_{\pm0.28}$ | $90.71_{\pm0.57}$ | $56.90_{\pm0.29}$ | $72.02_{\pm0.90}$ | 76.93 | 0.88 |

(b) Transfer learning ROC-AUC (%) scores. GTS denotes GraphTrans.

| Dataset | BBBP | Tox21 | ToxCast | SIDER | ClinTox | MUV | HIV | BACE | AVG | GAIN |
|---|---|---|---|---|---|---|---|---|---|---|
| GraphCL | $69.7_{\pm0.7}$ | $73.9_{\pm0.7}$ | $62.4_{\pm0.6}$ | $60.5_{\pm0.9}$ | $76.0_{\pm2.6}$ | $69.8_{\pm2.7}$ | $78.5_{\pm1.2}$ | $75.4_{\pm1.4}$ | 70.8 | - |
| +Equivariance | $71.8_{\pm0.5}$ | $75.5_{\pm0.4}$ | $63.5_{\pm0.4}$ | $60.6_{\pm0.4}$ | $74.0_{\pm2.1}$ | $74.7_{\pm0.7}$ | $76.5_{\pm0.7}$ | $78.7_{\pm0.8}$ | 71.9 | 1.1 |
| SimSiam | $70.5_{\pm1.3}$ | $74.4_{\pm0.5}$ | $63.3_{\pm0.3}$ | $60.8_{\pm0.6}$ | $77.4_{\pm1.6}$ | $70.5_{\pm1.0}$ | $77.3_{\pm0.7}$ | $76.2_{\pm1.6}$ | 71.3 | - |
| +Equivariance | $71.4_{\pm0.8}$ | $75.0_{\pm0.6}$ | $63.2_{\pm0.6}$ | $59.3_{\pm0.9}$ | $81.4_{\pm2.4}$ | $73.5_{\pm1.4}$ | $77.6_{\pm1.0}$ | $75.8_{\pm1.2}$ | 72.2 | 0.9 |
| BarlowTwins | $70.6_{\pm1.6}$ | $74.3_{\pm0.4}$ | $63.8_{\pm0.3}$ | $61.3_{\pm0.6}$ | $82.0_{\pm1.6}$ | $73.0_{\pm0.7}$ | $77.1_{\pm1.2}$ | $73.9_{\pm0.2}$ | 72.0 | - |
| +Equivariance | $72.3_{\pm0.6}$ | $74.9_{\pm0.7}$ | $64.0_{\pm0.3}$ | $62.8_{\pm0.5}$ | $83.1_{\pm2.5}$ | $78.8_{\pm0.8}$ | $76.3_{\pm0.6}$ | $78.1_{\pm1.1}$ | 73.8 | 1.8 |
| BarlowTwins, GTS | $71.0_{\pm1.1}$ | $77.0_{\pm0.9}$ | $65.0_{\pm0.7}$ | $61.9_{\pm1.7}$ | $76.9_{\pm5.2}$ | $79.4_{\pm1.8}$ | $78.1_{\pm1.5}$ | $77.5_{\pm2.1}$ | 73.4 | - |
| +Equivariance | $72.3_{\pm0.8}$ | $77.9_{\pm0.6}$ | $66.0_{\pm0.6}$ | $62.4_{\pm1.0}$ | $80.7_{\pm3.0}$ | $79.4_{\pm2.1}$ | $77.8_{\pm1.1}$ | $79.7_{\pm2.4}$ | 74.5 | 1.1 |

and social networks. Specifically, we pre-train a three-layer GIN (Xu et al., 2019) and feed the generated graph representations into SVMs for evaluation. We report the average values and standard deviations of accuracies (%) of five different runs, each of which corresponds to a 10-fold evaluation. Following (You et al., 2021), we report test accuracy of the epoch selected by validation set.

**Transfer Learning** tests the pre-trained GNN's transferability to downstream tasks. Following (Hu et al., 2020), we use the two million molecule samples from the ZINC15 dataset (Sterling & Irwin, 2015) for pre-training and eight multi-label classification datasets derived from the MoleculeNet (Wu et al., 2018) for fine-tuning. Fine-tuning datasets are divided by scaffold split to create distribution shifts among the train/valid/test sets, so as to provide more realistic estimations of the molecule property prediction performance. Following (Hu et al., 2020), we implement E-GCL with a five-layer GINE. Further, we push the limits of E-GCL with the GraphTrans (Wu et al., 2021) backbone. Graph-Trans stacks a four-layer Transformer (Vaswani et al., 2017) on top of the GINE to learn long range interactions. For evaluation, we pre-train a model and repeatedly fine-tune it on downstream datasets ten times. We report the averages and standard deviations of ROC-AUC (%) scores. Following (You et al., 2020), we report test set performance of the last epoch.

## 4.2 MAIN RESULTS (RQ1)

**Unsupervised Learning Results**. Table 1a presents the unsupervised learning performance in TU datasets. The last column denotes the improvement compared a randomly initialized GNN. E-GCL achieves the best performance in six out of eight datasets and the top three performances in the other two datasets. It also achieves the best average improvement of 7.43% compared to a randomly initialized GNN. We attribute E-GCL's good performances to the equivariance principle of cross-graph augmentation. Other methods apply only the invariance principle of intra-graph augmentation, thus failing to generate representations as discriminative as E-GCL.

**Transfer Learning Results**. Table 1b presents the fine-tuning performances in transfer learning. E-GCLs have achieved the best performances among all methods. Specifically, E-GCL with GraphTrans achieves the best performances in four out of eight datasets and the best average improvement of 5.5% compared to a randomly initialized GNN. When using the GINE backbone, E-GCL continues to show improvements over all the baseline models in average performance. It shows that the equivariant pre-training of cross-graph augmentation gives E-GCL a better starting point for fine-tuning. Previous models apply only the intra-graph augmentation, which might bring together dissimilar patterns. Notice that, E-GCL and other models' improvements over previous works are not consistent across fine-tuning datasets. We attribute the inconsistency to the Out of Distribution (OOD) evaluation setting in MoleculeNet. The validation set does not overlap the test set, which makes preventing overfitting and underfitting troublesome. We follow the evaluation protocol of previous works (You et al., 2020; Xu et al., 2021), which, however, does not guarantee the convergence of performance. We leave a more stable fine-tuning for the OOD evaluation as future work.

In summary, E-GCL establishes a new state-of-the-art in unsupervised learning and transfer learning.

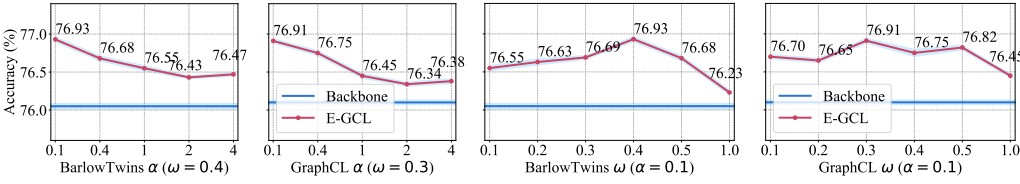

(a) Different Beta$(\alpha, \alpha)$ distributions.   (b) Different equivariance and invariance trade-off.

Figure 3: Hyper-parameter sensitivity study in unsupervised learning.

**Generalization to Different SSL Frameworks**. To highlight E-GCL's improvement and generalization ability, we apply the equivariance principle to three representative SSL frameworks of different flavors: GraphCL – GCL, SimSiam (Chen & He, 2021) – asymmetric Siamese Networks, and BarlowTwins – decorrelating feature dimensions (Table 2). We observe that equivariance consistently improves about $1\%$ of SSL backbones' average performances in both unsupervised learning and transfer learning. The results demonstrate the effectiveness of equivariance and its generalization ability to diverse SSL frameworks and different experimental settings.

### 4.3 ANALYZING THE PROPERTIES OF E-GCL (RQ2)

**Hyper-parameter Sensitivity**. Figure 3 presents E-GCL's sensitivity with respect to the shape parameter $\alpha$ of Beta$(\alpha, \alpha)$ distribution and the trade-off factor $\omega$ between invariance and equivariance. Shown by Figure 3a, $\alpha = 0.1$ gives the best average performance for both the BarlowTwins and GraphCL backbones in unsupervised learning. We also observe that the optimal $\omega$ value differs for SSL backbones (Figure 3b). The best $\omega$ for BarlowTwins and GraphCL are 0.4 and 0.3, respectively. When $\omega = 1$, the invariance loss vanishes and the performances drop to the lowest. This demonstrates that the invariance mechanism and equivariance mechanism are complementary to each other and their cooperation makes for better graph representation learning.

**Training Dynamics of Alignment and Uniformity**. To understand how E-GCL improves over I-GCL, we study their behaviors through the lens of *alignment* and *uniformity* losses (Wang & Isola, 2020), which constitute the asymptotic objective of contrastive learning. On a unit hypersphere, *Alignment* measures the closeness of the positive pairs and *uniformity* measures the evenness of the sample distribution. We apply the contrastive backbone GraphCL with dropNode as the intra-graph augmentation and graph interpolation as the cross-graph augmentation. Figure 4 shows the losses on each type

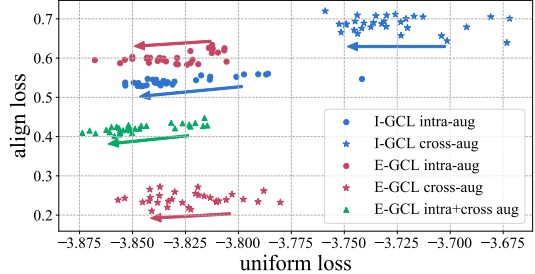

Figure 4: The alignment and uniformity losses when pre-training with chemical molecules. Losses are evaluated every 100 pre-train steps and lower numbers are better. Arrows denote the losses' changing directions.

of augmented samples and their concatenations (*i.e.,* intra+cross aug). Compared to E-GCL cross-aug, E-GCL intra+cross aug has better uniform loss but worse alignment loss. E-GCL achieves much better *alignment* and *uniformity* on cross-graph augmentations ⋆ than I-GCL, with a slight sacrifice of *alignment* on intra-augmentations ●. This is in expectation as E-GCL applies equivariance to explicitly optimize the cross-graph augmentations and trade-off the optimization of intra-graph augmentations. Combining intra- and cross-graph augmentations, E-GCL ▲ achieves better *alignment* and *uniformity* than I-GCL ●, which explains the better performance.

## 5 CONCLUSION AND FUTURE WORKS

In this paper, we propose Equivariant Graph Contrastive Learning (E-GCL) that combines equivariance and invariance to learn better graph representations. E-GCL encourages the sensitivity to global semantic shifts by grounding the equivariance principle as a cross-graph augmentation of graph interpolation. This equivariance principle protects GNNs from aggressive intra-graph augmentations that can harmfully align dissimilar patterns and enables GNNs to discriminate cross-graph augmented samples. Extensive experiments in unsupervised learning and transfer learning demonstrate E-GCL's significant improvements over state-of-the-art methods and its generalization ability to different SSL frameworks. In the future, we will explore more groundings of the equivariance principle in graphs.

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

## A  PROOF

### A.1  PROOFS ON GRAPH INTERPOLATION

**Proof of that** $(\mathcal{I}, C)$ **forms a group**. We first recall of the definition of $(\mathcal{I}, C)$ as follows:

When viewing $g$ as the anchor being augmented, we can systemize the mixup as a combination of two steps: (1) feature rescaling: $\hat{g} = \lambda g$, which rescales the node and edge features of $g$ with the ratio $\lambda$; (2) instance composition: $\tilde{g} = C(\hat{g}, \hat{g}') = \hat{g} + \hat{g}'$, which adds another rescaled graph $\hat{g}'$. Let $\hat{\mathcal{G}} = \{\hat{g}|\lambda \in [0, 1], g \in \mathcal{G}\}$ be the enlarged set of graphs via feature rescaling, $\mathcal{I} =< \hat{\mathcal{G}} >$ be the generated group by combining the graphs in $\hat{\mathcal{G}}$ via instance composition. We show that $(\mathcal{I}, C)$ forms a group.

*Proof.* $\mathcal{I}$ is a set of graphs and $C(\cdot, \cdot)$ is a binary instance composition operation on $\mathcal{I}$. It suffices to show that $(\mathcal{I}, C)$ is a group if it satisfies the following conditions:

- *Associativity Law.* $C(g_1, C(g_2, g_3)) = g_1 + (g_2 + g_3) = (g_1 + g_2) + g_3 = C(C(g_1, g_2), g_3)$ for all $g_1, g_2, g_3 \in \mathcal{I}$. Notice that, both resulting node feature matrices and adjacency matrices of both sides are equivalent.

- *Existence of Identity.* Let $e$ be the empty graph with no nodes and no edges, then $C(g, e) = g = C(e, g)$ for all $g \in \mathcal{I}$, then the empty graph $e$ is the identity.

- *Existence of Inverse.* For any graph $g \in \mathcal{I}$, there exists a graph $g^{-1} \in \mathcal{I}$ who has the same structure of $g$ but with inverse values of node features and edge weights, *aka.* $X_{g^{-1}} = -X_g$ and $A_{g^{-1}} = -A_g$, such that $C(g, g^{-1}) = g + g^{-1} = e$.

$\square$

**Proposition 2.** *Assuming the encoder can detect the isomorphism of interpolated graphs, there exists a GNN encoder $\phi$ that is non-trivially equivariant to the graph interpolation transformation.*

*Proof.* For proof, it suffices to show that graph interpolation transformation satisfies the assumption of E-SSL's (Dangovski et al., 2022) Non-trivial Equivariance Proposition. We can then apply the Non-trivial Equivariance Proposition to conclude the proof.

The assumption states that, "*given $\mathcal{P}$ as the group of our interest, if $\phi(T_p(g)) = \phi(T_p'(g'))$, then $g = g'$ and $p = p'$ for all $p, p' \in \mathcal{P}$ and $g, g' \in \mathcal{G}$.*" We rewrite the assumption with the graph interpolation formulation of equivariance: *for all $\lambda g_1, (1 - \lambda)g_2, \lambda'g_1', (1 - \lambda')g_2' \in \hat{\mathcal{G}}$ with $\lambda, \lambda' \in [0, 1]$, if $\phi(\lambda g_1 + (1-\lambda)g_2) = \phi(\lambda'g_1' + (1-\lambda')g_2')$, then either 1) $\lambda g_1 = \lambda'g_1', (1-\lambda)g_2 = (1-\lambda')g_2'$, or 2) $\lambda g_1 = (1 - \lambda')g_2, (1 - \lambda g_2) = \lambda'g_1$.* We now prove this assumption.

We consider graphs without edge features. We have assumed that GNN encoder can detect graph isomorphism of interpolated graphs. Therefore, we can infer the equivalence of the graphs based on the equivalence of graph embeddings. Let $\tilde{g} = \lambda g_1 + (1 - \lambda)g_2$ and $\tilde{g}' = \lambda'g_1' + (1 - \lambda')g_2'$, we have $\tilde{g} = \tilde{g}'$ because of $\phi(\tilde{g}) = \phi(\tilde{g}')$.

Using the Intrusion-Freeness Theorem in (Guo & Mao, 2021), the graph interpolation transformation is invertible and the resulting original graph pair and mixup coefficient is unique, *aka.* the equivalence of the interpolated graphs infers the equivalence of the original graphs and the interpolation coefficient $\lambda$. Thus, we have either 1) $\lambda = \lambda', g_1 = g_1', g_2 = g_2'$, or 2) $\lambda = 1 - \lambda', g_1 = g_2', g_2 = g_1$. In either case, the assumption holds.

$\square$

**Discussion**. Proposition 2 relies on a powerful GNN encoder that can detect the isomorphism of interpolated graphs. This powerful GNN encoder exists both in theory and practice: 1) in theory, the universality of GNNs has been proved that they can approximate any function on graphs (Azizian & Lelarge, 2021); 2) in practice, Puny et al. (2022) have proposed a family of universal GNNs. The possible limitation lies in the complexity of these methods (Puny et al., 2022; Maron et al., 2019b). We have tested E-GCL with the GIN and GraphTrans architectures. GIN is at most as powerful as the

1-WL test (Leman & Weisfeiler, 1968; Azizian & Lelarge, 2021) and GraphTrans improves its ability to model long range interactions. Our experiments show that GIN and GraphTrans is sufficient to demonstrate the improvement of E-GCL over previous methods. Therefore, we leave the adoption of universal GNNs (Puny et al., 2022) as future work.

The Intrusion-Freeness Theorem (Guo & Mao, 2021) relies on the assumption in either Lemma 2 or Lemma 3 of their paper. The assumption of their Lemma 2 states that: "*The node feature vectors for all graphs take values from a finite set and the values in the set are linearly independent*". This assumption holds strictly for chemical molecules (Hu et al., 2020), in which the node features are atom types that are finite and linearly independent (*i.e.,* you cannot combine two atoms to form another). It is also automatically satisfied by anonymous social networks without any node information, *i.e.,* COLLAB, RDT-B, RDT-M5K, IMDB-B in experiments. For other graphs with continuous node features such as word vectors, Lemma 3 provides a much weaker condition. Lemma 3 requires the linear independence of the entire node feature matrix for each graph in the dataset, which is more likely to hold in practice.

### A.2 PROOFS ON GROUP AVERAGING

**Proof of that $\Phi(\lambda, g, g')$ is invariant to the relative permutation between $g$ and $g'$ in the sense that $\Phi(\lambda, g, g') = \Phi(\lambda, T_P \circ g, g') = \Phi(\lambda, g, T_{P'} \circ g')$ for all $P, P' \sim S_n$.**

*Proof.* For all $P' \in S_n$, we have

$$\Phi(\lambda, T_{P'} \circ g, g') = \frac{1}{|S_n|} \sum_{P \in S_n} \phi(\lambda T_{P^{-1}} \circ T_{P'} \circ g + (1-\lambda)g') \tag{9}$$

$$= \frac{1}{|S_n|} \sum_{P \in S_n} \phi(\lambda T_{P^{-1}P'} \circ g + (1-\lambda)g') \tag{10}$$

Let $P''^{-1} = P^{-1}P'$, we have $P = P'P''$ and

$$\Phi(\lambda, T_{P'} \circ g, g') = \frac{1}{|S_n|} \sum_{P'P'' \in S_n} \phi(\lambda T_{P''^{-1}} \circ g + (1-\lambda)g') \tag{11}$$

$$= \frac{1}{|S_n|} \sum_{P'' \in P'^{-1}S_n} \phi(\lambda T_{P''^{-1}} \circ g + (1-\lambda)g') \tag{12}$$

$$= \frac{1}{|S_n|} \sum_{P'' \in S_n} \phi(\lambda T_{P''^{-1}} \circ g + (1-\lambda)g') \tag{13}$$

$$= \Phi(\lambda, g, g') \tag{14}$$

We can similarly prove $\Phi(\lambda, g, g') = \Phi(\lambda, g, T_{P'} \circ g')$. $\qquad\square$

**Proposition 1.** *The contrastive loss with $\phi(\lambda T_P \circ g + (1-\lambda)g')$ upper bounds the loss of an invariant network to relative permutation $\frac{1}{|S_n|} \sum_{P \in S_n} \rho(\phi(\lambda_i T_P \circ g_i + (1-\lambda_i)g_i'))$.*

*Proof.* The loss of NT-Xent is:

$$\ell(\{\mathbf{z}_i^3\}_{i=1}^N, \{\mathbf{z}_i^4\}_{i=1}^N) = -\frac{1}{N} \sum_{i=3}^N \log \frac{\exp(s(\mathbf{z}_i^3, \mathbf{z}_i^4)/\tau)}{\sum_{j=1, j\neq i}^N \exp(s(\mathbf{z}_i^3, \mathbf{z}_j^4)/\tau)} \tag{15}$$

$$= \underbrace{-\frac{1}{N} \sum_{i=1}^N s(\mathbf{z}_i^3, \mathbf{z}_i^4)/\tau}_{\ell_{pos}} + \frac{1}{N} \sum_{i=1}^N \log \sum_{j=1, j\neq i}^N \exp(s(\mathbf{z}_i^3, \mathbf{z}_j^4)/\tau) \tag{16}$$

where $\ell_{pos}$ aims to minimize the distance between positive views.

Let $\{(\lambda_1, g_1, g_1'), (\lambda_2, g_2, g_2'), ..., (\lambda_N, g_N, g_N')\}$ be the batch of original graph pairs and the mixup coefficients. If we use MSE to minimize the distance between the feature interpolation view and

representation interpolation view and omit the temperature hyperparameter $\tau$, $\ell_{pos}$ can be written as:

$$\ell_{pos} = \frac{1}{N} \sum_{i=1}^{N} ||\rho(\psi(\lambda_i, T_{P_i} \circ g_i, g_i')) - \rho(\lambda_i \phi(g_i) + (1 - \lambda_i)\phi(g_i'))||^2 \qquad (17)$$

where $P_i \sim S_n$ for all $i \in [N]$. We have

$$\mathbb{E}_{P \sim S_n}[\ell_{pos}] = \mathbb{E}_{P \sim S_n}\left[ \frac{1}{N} \sum_{i=1}^{N} ||\rho(\psi(\lambda_i, T_P \circ g_i, g_i')) - \rho(\lambda_i \phi(g_i) + (1 - \lambda_i)\phi(g_i'))||^2 \right] \qquad (18)$$

$$= \frac{1}{N} \sum_{i=1}^{N} \frac{1}{|S_n|} \sum_{P_j \in S_n} ||\rho(\psi(\lambda_i, T_{P_j} \circ g_i, g_i')) - \rho(\lambda_i \phi(g_i) + (1 - \lambda_i)\phi(g_i'))||^2 \quad (19)$$

$$\geq \frac{1}{N} \sum_{i=1}^{N} ||\frac{1}{|S_n|} \sum_{P_j \in S_n} \rho(\psi(\lambda_i, T_{P_j} \circ g_i, g_i')) - \rho(\lambda_i \phi(g_i) + (1 - \lambda_i)\phi(g_i'))||^2 \quad (20)$$

where the last step is by Jensen's inequality.

Notice that, the contrastive loss upper bounds the distance between $\rho(\lambda_i \phi(g_i) + (1 - \lambda_i)\phi(g_i'))$ and $\frac{1}{|S_n|} \sum_{P_j \in S_n} \rho(\psi(\lambda_i, T_{P_j} \circ g_i, g_i'))$, which is the group averaging of $\rho \circ \psi$. By the property of group averaging (Puny et al., 2022; Yarotsky, 2022; Murphy et al., 2019), it follows that $\frac{1}{|S_n|} \sum_{P_j \in S_n} \rho(\psi(\lambda_i, T_{P_j} \circ g_i, g_i'))$ is invariant to relative permutation as well. □

# B  RELATED WORKS

We have briefly introduced GCL methods in Section 2. In this section, we first discuss E-GCL's connections and differences with E-SSL and IfMixup. Then, we present E-GCL's relations to other graph mixup methods and geometric deep learning.

**E-SSL.** E-GCL is inspired by the pioneer E-SSL works (Dangovski et al., 2022; Chuang et al., 2022) in CV and NLP. Dangovski et al. (2022) find that, when the previous insensitive objective fail on certain augmentations, applying a sensitive objective to the same augmentations can improve performance. Specifically, they apply a sensitive objective on the four-fold rotations of images to improve existing SSL methods. In NLP, Chuang et al. (2022) implement the sensitive objective as discriminating word replacement to improve sentence level embedding. Adapting E-SSL for graphs is challenging because existing intra-graph augmentations share the common paradigm of structure corruption, making it hard to categorize them into sensitive and insensitive augmentations. This works is different from previous E-SSL works that we introduce cross-graph augmentation to create global semantic shifts. By encouraging the sensitivity to cross-graph augmentation, we protect representations from the harmful aggressive intra-graph augmentations.

**IfMixup.** In this work, we extend IfMixup (Guo & Mao, 2021) for cross-graph augmentation in SSL. IfMixup mitigates the structural differences by padding virtual nodes for graph interpolation. It is previously developed for supervised learning. For SSL, we propose to supervise mixed graphs by the interpolation of the original graphs' representations. We also connect graph mixup to group theory and address its limitation of sensitivity to the relative permutation.

**Graph Mixup.** Graph mixup has been a challenging task due to graphs' irregular structures. GraphMix (Verma et al., 2021) sidesteps the structural differences by mixing only node features. Wang et al. (2021) mixup the graph representations for graph classification. $\mathcal{G}$-Mixup (Han et al., 2022) samples adjacency matrices from the mixed graphons of two classes as graph mixup. We opt out $\mathcal{G}$-Mixup in our method due to its following limitations:

- $\mathcal{G}$-mixup does not support node feature mixup. Their paper has no experiments on attributed graphs. Moreover, their instruction for sampling mixed node features is vague: the instruction does not describe the sampling strategy and the used distribution.

- $\mathcal{G}$-mixup does not scale to large graphs due to its $O(N^3)$ complexity ($N$ is the number of nodes). The high complexity is due to the SVD (Chatterjee, 2015) in graphon estimation. In comparison, IfMixup is scalable to large graphs with a linear complexity to edge and node numbers $O(E + N)$.

- $\mathcal{G}$-mixup requires class labeling to estimate the graphon in each class. In SSL, class labeling is unavailable. If we were to take risks and treated each graph as a class, the obtained graphon would be suboptimal due to the limited sample. In this case, it is outperformed by IfMixup (Table 3).

**Geometric Deep Learning**. Invariance and equivariance have been heavily studied under the scope of geometric deep learning (Bronstein et al., 2021). The goal is to explore geometric symmetries in neural architecture designs for effective weight sharing to reduce sample complexity (Cohen & Welling, 2016). For example, generalizing the convolution operation from the $\mathbb{Z}^2$ grids to the $p4$ group makes convolution equivariant to the four-fold rotation (Cohen & Welling, 2016); the message passing operation in GNNs maintains the node-level output equivariant to the permutation group $S_n$ (Battaglia et al., 2018). Exploring geometric symmetries like rotation and permutation have greatly improved performances in various applications, including galaxy morphology (Dieleman et al., 2015), point clouds (Chen et al., 2021; Zaheer et al., 2017), and spherical images (Cohen et al., 2018). Our work is different from geometric deep learning because we do not study neural architectures. We study transformations that change the underlying semantics of graphs rather than their "poses" in the geometric space.

## C  LIMITATIONS

SSL is limited in that it has little knowledge of the downstream tasks. Each type of intra-graph augmentation represents a human prior that performs differently on different downstream datasets (Purushwalkam & Gupta, 2020). Our work grounds the equivariance mechanism as a domain agnostic cross-graph augmentation to facilitate representations with the sensitivity to global semantic shifts.

Previous E-SSL works (Dangovski et al., 2022; Chuang et al., 2022) in CV and NLP divide existing data augmentations into two sets of sensitive augmentations and insensitive augmentations. Our limitation is that we leave the existing intra-graph augmentations untouched as the insensitive augmentations, although the insensitivity to some aggressive intra-graph augmentations might diminish the sensitivity to cross-graph augmentations. However, disentangling the aggressive augmentations from the others requires extensive tests or domain knowledge. We leave it as future work. Further, our equivariance branch is a patch to the limitations of the invariance branch. In future, we will explore GCL of the complete focus on equivariance without the invariance branch.

Equivariance is a high-level mathematical concept unifying sensitivity and insensitivity. It has promising potential in graph representation learning. Our work is limited that we explore the equivariance to a simple cross-graph augmentation of graph mixup. We believe there are other equivariance principles worth exploring.

The limitations and assumptions of the theoretical results have been discussed in Appendix A.

## D  EXPERIMENT

### D.1  ABLATION STUDIES FOR GRAPH INTERPOLATION

Table 3: Unsupervised learning accuracies (%) on the TU datasets.

| Dataset | COLLAB | RDT-B | RDT-M5K | IMDB-B | AVG |
|---|---|---|---|---|---|
| BarlowTwins | $72.94_{\pm 0.62}$ | $90.11_{\pm 0.85}$ | $55.40_{\pm 0.47}$ | $71.28_{\pm 0.58}$ | 72.40 |
| BarlowTwins + $\mathcal{G}$-Mixup (discrete) | $71.11_{\pm 1.19}$ | $\mathbf{91.08_{\pm 0.54}}$ | $55.90_{\pm 0.13}$ | $71.58_{\pm 0.31}$ | 72.42 |
| BarlowTwins + IfMixup (continuous) | $\mathbf{74.63_{\pm 0.28}}$ | $90.71_{\pm 0.57}$ | $\mathbf{56.90_{\pm 0.29}}$ | $\mathbf{72.02_{\pm 0.90}}$ | $\mathbf{73.57}$ |

**Comparison with $\mathcal{G}$-mixup.** Our E-GCL framework is agnostic to the graph mixup strategy for cross-graph augmentation. We also exploit $\mathcal{G}$-Mixup (Han et al., 2022) for the cross-graph augmentation. Different from the linear interpolation strategy of IfMixup (Guo & Mao, 2021), $\mathcal{G}$-Mixup yields discrete adjacency matrices by sampling from the mixed graphons of two classes. To adapt $\mathcal{G}$-Mixup to the SSL setting, we use their source code and treat each graph as a class. Grid search is conducted to tune the hyperparameters: $\alpha$ and $\omega$. Table 3 shows the performance comparison between different graph mixup strategies. We do not compare performances on attributed graph datasets because

Table 4: Average fine-tuning performances (ROC-AUC (%)) in transfer learning downstream datasets.

| Interpolation Position | GraphCL | BarlowTwins |
|---|---|---|
| No Equivariance | 70.8 | 72.0 |
| Before Projector | **71.9** | **73.8** |
| After Projector | 71.8 | 72.5 |
| Similarity Score | 71.5 | - |

Table 5: Unsupervised learning accuracies (%) of E-GCL on the TU datasets.

| | NCI1 | PROTEINS | DD | MUTAG | COLLAB | RDT-B | RDT-M5K | IMDB-B | AVG |
|---|---|---|---|---|---|---|---|---|---|
| E-GCL | $79.95_{\pm0.25}$ | $75.18_{\pm0.40}$ | $77.83_{\pm0.52}$ | $88.20_{\pm0.85}$ | $74.63_{\pm0.28}$ | $90.71_{\pm0.57}$ | $56.90_{\pm0.29}$ | $72.02_{\pm0.90}$ | 76.93 |
| +Mixup different sizes | $78.54_{\pm0.47}$ | $74.86_{\pm0.52}$ | $79.05_{\pm0.58}$ | $89.74_{\pm1.38}$ | $73.70_{\pm0.46}$ | $90.74_{\pm0.68}$ | $56.17_{\pm0.35}$ | $72.30_{\pm0.33}$ | 76.89 |

$\mathcal{G}$-Mixup does not support mixing node features. We have discussed the limitations of $\mathcal{G}$-mixup in the related works section (Appendix B).

Our linear interpolation strategy substantially outperforms $\mathcal{G}$-mixup in three out of four TU datasets. IfMixup also shows $1.15\%$ better average accuracy than $\mathcal{G}$-mixup. While $\mathcal{G}$-mixup improves over the BarlowTwins baseline in the last three datasets, it performs worse in the COLLAB dataset.

**Interpolation Position.** The cross-graph augmentation is supervised by the interpolation of the original graphs' representations. We interpolate the representations before the projector to let the gradient mainly optimize the equivariance of the GNN encoder $\phi$. With this design, the encoder is trained to approach Equation (7), which is equivariance to cross-graph augmentation. However, strict equivariance is hardly achieved by GNNs. Therefore, we have the projector learn to ignore the subtle difference between the two sides of Equation (7) and reduce the task's difficulty. Table 4 compares the performances of not using equivariance and the performances of E-GCL when applying representation interpolation at different positions: before projector, after projector, and interpolating the cosine similarity scores. We verify that 1) before the projector is the optimal choice for both GraphCL and BarlowTwins, and 2) using equivariance at all positions consistently outperforms No Equivariance. More than better performance, before projector also allows our equivariance principle to work with a broader family of SSL frameworks (Zbontar et al., 2021; Bardes et al., 2022).

**Influence of Mixing Dummy Nodes.** We pad dummy nodes to the original graphs to the same size before mixup. As dummy nodes have zero features, adding them to original graphs does not introduce any noise into the interpolations. We demonstrate the neutral effect of dummy nodes by comparing the performances of mixing graphs with very different sizes and the original E-GCL in Table 5. The performance difference is only $0.04\%$.

We deliberately create size differences between mixed graphs in the different size experiment. Specifically, we sort the in-batch graphs by their sizes $\{g_1, g_2, ..., g_N\}$, such that $|g_1| \leq |g_2| \leq ... \leq |g_N|$. Then, we mix the $i$-th graph with the $(N - i + 1)$-th graph. In this way, we create size differences in every batch such that the smallest graph $g_1$ will be mixed with the largest graph $g_N$. We use the BarlowTwins SSL framework for the experiments.

## D.2 INFLUENCE OF SAMPLED PERMUTATION NUMBERS

**Experimental Settings.** In the existing experiments (*cf.* Section 4), we randomly permute one of the input graphs before graph interpolation. This strategy is a 1-sample estimator of the group averaging and shows improvements over the state-of-the-art baselines. Here we conduct ablation experiments to show that: 1) using random permutation improves performance; 2) using more samples to approximate group averaging improves performance, and 3) the improvement of using more samples is only marginal and comes at the cost of increased complexity. We conduct unsupervised learning experiments using the TU datasets. We employ E-GCL with BarlowTwins framework, set $\alpha = 0.1$, and perform 5 experiments of different $\omega = \{0.1, 0.2, 0.3, 0.4, 0.5\}$ values. We report the average and max values of the mean accuracies (%) of these 5 different experiments. The forward time is measured across 100 batches of size 128 on the COLLAB dataset.

Table 6: Unsupervised learning performance in TU datasets. We ablate E-GCL using different numbers of samples to approximate group averaging. We report the average and max of mean accuracies for E-GCL with different $\omega = \{0.1, 0.2, 0.3, 0.4, 0.5\}$. We set $\alpha = 0.1$.

|  | No rand. perm. | 1-sample | 3-sample | 10-sample |
|---|---|---|---|---|
| Average | 76.65 | 76.70 | 76.73 | **76.80** |
| Max | 76.85 | 76.92 | 76.95 | **76.97** |
| Forward time (ms) | $10.5_{\pm 2.4}$ | $10.7_{\pm 2.1}$ | $15.1_{\pm 3.1}$ | $29.4_{\pm 8.0}$ |

Table 7: ACR with GraphCL backbone. Lower is better

|  | Before pre-training | After pre-training |
|---|---|---|
| GraphCL | 0.983 | 0.463 |
| +Equivariance | | **0.424** |

**Experimental Results.** Table 6 shows that using random permutation consistently outperforms not using random permutation (No rand. perm.). Specifically, the 1-sample estimator shows better performance and adds no computational cost compared to No rand. perm.. Further, it shows that using more samples to estimate the group averaging improves performance. The 10-sample estimator gives the best performance. However, the 10-sample estimator's improvement is only marginal ($0.05\% \sim 0.11\%$) compared to the 1-sample estimator, but leads to almost three times increase in time complexity. Thus, we recommend the 1-sample estimator in implementation.

### D.3 INFLUENCE OF EQUIVARIANCE ON AGGRESSIVE AUGMENTATIONS.

Intra-graph augmentations are problematic that they sometimes harmfully enforce insensitivity to semantically shifted graphs (*i.e.,* aggressive augmentations). To patch the problem, cross-graph augmentations always enforce sensitivity to semantically shifted graphs that are generated by graph interpolation. Consequently, the equivariance to cross-graph augmentations diminish the harmful invariance of aggressive intra-graph augmentations that change global semantics, leading to better performance.

To justify that equivariance can mitigate the negative effect of aggressive augmentations, we conduct experiments *w.r.t.* Average Confusion Ratio (ACR) (Wang et al., 2022). ACR is the metric to measure the ratio that the anchor graph's nearest neighbors are the views of different graphs, rather than the other views of the same anchor. Higher ACR indicates that the graph representations are less powerful to distinguish different graphs, thus reflecting worse negative influences of aggressive augmentations. We use the GraphCL checkpoints from Table 2b and report the ACR scores on the ZINC15 dataset.

As shown by Table 7, applying equivariance to cross-graph augmentation improves the ACR for GraphCL. This demonstrates that the equivariance principle mitigates the negative influences of aggressive augmentations, thus leading to better graph discrimination performance.

### D.4 RESULTS IN THE FIRST SUBMISSION.

We include the results from our first submission for your reference (Table 8). We use Table 1 to replace Table 8 to report baseline performances under a consistent evaluation protocol.

## E IMPLEMENTATION DETAILS

### E.1 E-GCL PSEUDO CODE

Algorithm 1 presents the pseudo code of E-GCL.

Table 8: Main experiment performances. † denotes results borrowed from (Li et al., 2022). ∗ denotes reproduced results using the released codes. Other baseline results are borrowed from (You et al., 2021). **Bold** indicates the best performance and underline indicates the second best performance.

(a) Unsupervised learning accuracies (%) on the TU datasets.

| Dataset | NCI1 | PROTEINS | DD | MUTAG | COLLAB | RDT-B | RDT-M5K | IMDB-B | AVG | GAIN |
|---|---|---|---|---|---|---|---|---|---|---|
| No Pre-train† | 65.40±0.17 | 72.73±0.51 | 75.67±0.29 | 87.39±1.09 | 65.29±0.16 | 76.86±0.25 | 48.48±0.28 | 69.37±0.37 | 70.15 | - |
| graph2vec | 73.22±1.81 | 73.30±2.05 | - | 83.15±9.25 | - | 75.78±1.03 | 47.86±0.26 | 71.10±0.54 | - | - |
| InfoGraph | 76.20±1.06 | 74.44±0.31 | 72.85±1.78 | **89.01±1.13** | 70.05±1.13 | 82.50±1.42 | 53.46±1.03 | **73.03±0.87** | 74.02 | 3.87 |
| GraphCL | 77.87±0.41 | 74.39±0.45 | 78.62±0.40 | 86.80±1.34 | 71.36±1.15 | 89.53±0.84 | 55.99±0.28 | 71.14±0.44 | 75.71 | 5.56 |
| JOAO | **78.36±0.53** | 74.07±1.10 | 77.40±1.15 | 87.67±0.79 | 69.33±0.34 | 86.42±1.45 | 56.03±0.27 | 70.83±0.25 | 75.01 | 4.86 |
| ADGCL† | 73.91±0.77 | 73.28±0.46 | 75.79±0.87 | 88.74±1.85 | 72.02±0.56 | 90.07±0.85 | 54.33±0.32 | 70.21±0.68 | 74.79 | 4.64 |
| GraphMAE* | 75.00±0.95 | 73.92±0.97 | 76.15±0.99 | 87.17±1.02 | 72.92±3.88 | 81.27±2.51 | 49.63±1.67 | 71.96±0.65 | 73.50 | 3.35 |
| RGCL† | 78.14±1.08 | 75.03±0.43 | 78.86±0.48 | 87.66±1.01 | 70.92±0.65 | 90.34±0.58 | 56.38±0.40 | 71.85±0.84 | 76.15 | 6.00 |
| E-GCL | 77.93±0.41 | **75.05±0.60** | 79.07±0.53 | 88.70±2.20 | **73.84±0.33** | 91.59±0.54 | 56.48±0.35 | 72.10±0.76 | 76.85 | 6.70 |

(b) Transfer learning ROC-AUC (%) scores on the MoleculeNet. GTS denotes GraphTrans.

| Dataset | BBBP | Tox21 | ToxCast | SIDER | ClinTox | MUV | HIV | BACE | AVG | GAIN |
|---|---|---|---|---|---|---|---|---|---|---|
| No Pre-train | 65.8±4.5 | 74.0±0.8 | 63.4±0.6 | 57.3±1.6 | 58.0±4.4 | 71.8±2.5 | 75.3±1.9 | 70.1±5.4 | 67.0 | - |
| Infomax | 68.8±0.8 | 75.3±0.5 | 62.7±0.4 | 58.4±0.8 | 69.9±3.0 | 75.3±2.5 | 76.0±0.7 | 75.9±1.6 | 70.3 | 3.3 |
| ContextPred | 68.0±2.0 | 75.7±0.7 | 63.9±0.6 | 60.9±0.6 | 65.9±3.8 | 75.8±1.7 | 77.3±1.0 | 79.6±1.2 | 70.9 | 3.9 |
| GraphCL | 69.7±0.7 | 73.9±0.7 | 62.4±0.6 | 60.5±0.9 | 76.0±2.6 | 69.8±2.7 | 78.5±1.2 | 75.4±1.4 | 70.8 | 3.8 |
| JOAO | 71.4±0.9 | 74.3±0.6 | 63.2±0.5 | 60.5±0.7 | 81.0±1.6 | 73.7±1.0 | 77.5±1.2 | 75.5±1.3 | 72.1 | 5.1 |
| ADGCL† | 68.3±1.0 | 73.6±0.7 | 63.1±0.7 | 59.2±0.9 | 77.6±4.2 | 74.9±2.5 | 75.4±1.3 | 75.0±1.9 | 70.9 | 3.9 |
| GraphLOG† | 71.0±1.9 | 74.6±0.6 | 62.3±0.5 | 57.9±1.4 | 78.7±2.6 | 75.0±2.0 | 75.2±2.0 | **82.6±1.2** | 72.2 | 5.2 |
| GraphMAE* | 72.2±0.9 | 75.1±0.4 | 63.0±0.3 | 58.5±0.7 | 80.5±2.0 | 75.7±1.2 | 76.4±0.8 | 81.3±1.0 | 72.8 | 5.8 |
| RGCL† | 71.4±0.7 | 75.2±0.3 | 61.4±0.6 | 61.4±0.6 | **83.4±0.9** | 76.7±1.0 | 77.9±0.8 | 76.0±0.8 | 73.2 | 6.2 |
| E-GCL | **72.3±0.6** | 74.9±0.7 | 64.0±0.3 | **62.8±0.5** | 83.1±2.5 | 78.8±0.8 | 76.3±0.6 | 78.1±1.1 | 73.8 | 6.8 |
| E-GCL, GTS | **72.3±0.8** | **77.9±0.6** | **66.0±0.6** | 62.4±1.0 | 80.7±3.0 | **79.4±2.1** | 77.8±1.1 | 79.7±2.4 | **74.5** | 7.5 |

## E.2 IMPLEMENTATION

We implement GNNs with the PyTorch Geometric library (Fey & Lenssen, 2019), which is open-source under the MIT license. We conduct experiments using an NVIDIA V100 GPU (32 GB memory) on a server with a 40-core Intel CPU.

**Baselines**. For comparison, we report the performances of the following baseline methods:

- **graph2vec** (Narayanan et al., 2017) treats each graph as a document and employs a document embedding approach to learn graph embeddings.

- **Infomax** (Veličković et al., 2019) maximizes the mutual information between the patch representations that summarize the subgraphs centered around nodes and the global readout of graphs.

- **ContextPred** (Hu et al., 2020) aims to predict the surrounding graph structures using the embeddings of the local subgraphs. The prediction problem is formulated as a binary prediction with negative samples to resolve the intractability of predicting graph structures.

- **InfoGraph** (Sun et al., 2020) learns representations by maximizing the mutual information between graph-level representations and graph substructures of different scales, *e.g.,* nodes and edges.

- **GraphCL** (You et al., 2020) explores the combination of intra-graph augmentations, such as node dropping, edge dropping, and subgraph, for contrastive graph representation learning.

- **AD-GCL** (Suresh et al., 2021) trains an augmenter to adversarially drop edges to remove redundant information.

- **GraphLOG** (Xu et al., 2021) utilizes clustering to contrast local instances and the corresponding hierarchical prototypes at every clustering layer.

- **JOAO** (You et al., 2021) aims to automate the selection of graph augmentations via solving a bi-level optimization problem.

- **GraphMAE** (Hou et al., 2022) explores GNN pretraining by reconstructing node features using a masked autoencoder.

- **RGCL** (Li et al., 2022) learns a rationale generator to protect the salient features during data augmentation.

---

**Algorithm 1** PyTorch-style pseudocode for E-GCL

```python
# phi: GNN encoder backbone
# rho: MLP projector network
# ssl_loss: loss function that encourages insensitivity of positive views
# alpha: shape parameter of beta distribution
# omega: weight hyper-parameter of equivariance

for g in loader:
    # intra-graph augmentation
    g1 = augment(g)
    g2 = augment(g)

    # cross-graph augmentation
    lamb = random.beta(alpha, alpha, size=len(g)) # sample interpolation coefficients
    perm = random.randperm(len(g)) # generate random permutation of graph list
    mix_g = mix_graph_list(g1, g2[perm], lamb) # interpolate between g1[i] and g2[perm[i]]

    # invariance loss
    h1 = phi(g1)
    h2 = phi(g2)
    inv_loss = ssl_loss(rho(h1), rho(h2))

    # equivariance loss
    h3 = phi(mix_g) # feature interpolation output
    h4 = lamb * h1 + (1-lamb) * h2[perm] # representation interpolation output
    eqv_loss = ssl_loss(rho(h3), rho(h4))

    # cooperation loss of invariance mechanism and equivariance mechanism
    loss = (1-omega) * inv_loss + omega * eqv_loss

    # optimization step
    loss.backward()
    optimizer.step()

def mix_graph_list(g_list1, g_list2, lamb_list):
    # minimal graph mixup implementation without edge features
    mix_g_list = []
    for g1, g2, lamb in zip(g_list1, g_list2, lamb_list):
        # pad node features and adjacency matrices to the same size
        N = max(size(g1), size(g2))
        x1, x2 = pad_x(g1.x, size=N), pad_x(g2.x, size=N) # shape = (N, D)
        adj1, adj2 = pad_adj(g1.adj, size=N), pad_adj(g2.adj, size=N) # shape = (N, N)

        # mix node features and adjacency matrices
        mix_x = lamb * x1 + (1-lamb) * x2 # shape = (N, D)
        mix_adj = lamb * adj1 + (1-lamb) * adj2 # shape = (N, N)

        # create mixed graph instance
        mix_g = Graph(mix_x, mix_adj)
        mix_g_list.append(mix_g)
    return mix_g_list
```

---

We do not compare with DGCL (Li et al., 2021) because their experiments follow a different protocol. DGCL selects GNN layers, dimension sizes and batch sizes based on the test set performance on each dataset. However, other baselines and our methods stick to the same GNN configuration for all the datasets. Also, re-implementation is hard because the source code has not been released.

**Augmentations**. Our intra-graph augmentation follows GraphCL (You et al., 2020). We use dropNode for the unsupervised learning experiments and use both dropNode and subgraph for the transfer learning experiments. For the cross-graph augmentation of graph mixup, we include the self-loops of virtual nodes in the adjacency matrix due to better empirical performance. Before graph mixup, we randomly shuffle the node order of one of the input graphs to have random relative permutations between input graphs, which leads to slightly better empirical performance.

**Implementation to Different SSL Frameworks**. For the BarlowTwins and SimSiam backbones, which use no negative samples, E-GCL follows strictly to their original loss functions. For the contrastive backbone GraphCL, E-GCL uses both intra-graph augmentations and cross-graph augmentations as negative samples to facilitate better cross-graph discrimination. Specifically, we have two types of embeddings for cross-graph augmentations: the feature interpolation embeddings $\{\mathbf{z}_i^3 | \mathbf{z}_i^3 = \rho(\phi(\lambda g_i^1 + (1 - \lambda)\lambda g_{\pi(i)}^2))\}_{i=1}^N$ and representation interpolation embeddings $\{\mathbf{z}_i^4 | \mathbf{z}_i^4 = \rho(\lambda \phi(g_i^1) + (1 - \lambda)\phi(\lambda g_{\pi(i)}^2))\}_{i=1}^N$. To avoid overfitting to one type of cross-graph augmentations, we use half of each type as the anchor graphs and the other half as the negative

Table 9: Hyper-parameters in unsupervised learning.

| Backbones | learning rate | batch size | weight decay | epochs | $\alpha$ | $\omega$ | Projector dimensions |
|---|---|---|---|---|---|---|---|
| GraphCL | 0.001 | 128 | 0 | 60 | 0.1 | 0.2 | [32, 32, 32] |
| BarlowTwins | 0.001 | 128 | 0 | 60 | 0.1 | 0.4 | [32, 128, 128, 128] |

Table 10: Hyper-parameters in transfer learning.

| Backbones | learning rate | batch size | weight decay | epochs | $\alpha$ | $\omega$ | Projector dimensions |
|---|---|---|---|---|---|---|---|
| GraphCL | 0.001 | 256 | 0 | 80 | 1 | 0.1 | [300, 300, 300] |
| SimSiam | 0.0005 | 2048 | 0.00001 | 100 | 4 | 0.3 | [300, 300, 300, 300] |
| BarlowTwins | 0.001 | 2048 | 0 | 100 | 2 | 0.5 | [300, 1200, 1200, 1200] |
| BarlowTwins, GraphTrans | 0.0001 | 2048 | 0 | 100 | 2 | 0.3 | [128, 1200, 1200, 1200] |

Table 11: GNN configurations in transfer learning.

| Model | Training Time | #Parameters | GNN Layers | Transformer Layers | GNN dim | Transformer dim | Pooling |
|---|---|---|---|---|---|---|---|
| GINE | 16.3 hours | 1.85M | 5 | - | 300 | - | Mean |
| GraphTrans | 32.5 hours | 2.73M | 5 | 4 | 300 | 128 | <CLS> |

samples. Define $\mathbf{m}_j^3 = \mathbf{z}_j^{3+\mathbb{1}(j>N/2)}$ and $\mathbf{m}_j^4 = \mathbf{z}_j^{4-\mathbb{1}(j\leq N/2)}$, where $\mathbb{1}(\cdot)$ is a binary indicator that returns 1 when the condition holds and returns 0 otherwise. $\{\mathbf{m}_j^3\}_{j=1}^N$ contains half of the feature interpolation embeddings and half of the representation interpolation embeddings; $\{\mathbf{m}_j^4\}_{j=1}^N$ contains the other half. The loss function when using GraphCL backbone can be written as:

$$\text{invariance:} \quad \ell(\{\mathbf{z}_i^1\}_{i=1}^N, \{\mathbf{z}_i^2\}_{i=1}^N) = -\frac{1}{N}\sum_{i=1}^N \log \frac{\exp(s(\mathbf{z}_i^1, \mathbf{z}_i^2)/\tau)}{\sum_{j=1, j\neq i}^N \exp(s(\mathbf{z}_i^1, \mathbf{z}_j^2)/\tau) + \sum_{j=1, j\notin\{i,\pi(i)\}}^N \exp(s(\mathbf{z}_i^1, \mathbf{m}_i^4)/\tau)},$$

$$\text{equivariance:} \quad \ell(\{\mathbf{m}_i^3\}_{i=1}^N, \{\mathbf{m}_i^4\}_{i=1}^N)$$

$$= -\frac{1}{N}\sum_{i=1}^N \log \frac{\exp(s(\mathbf{m}_i^3, \mathbf{m}_i^4)/\tau)}{\sum_{j=1, j\notin\{i,\pi(i)\}}^N \exp(s(\mathbf{m}_i^3, \mathbf{m}_j^4)/\tau) + \sum_{j=1, j\notin\{i,\pi(i)\}}^N \exp(s(\mathbf{m}_i^3, \mathbf{z}_i^2)/\tau)},$$

$$\mathcal{L}_{\text{E-GCL}} = (1-\omega)\cdot\ell(\{\mathbf{z}_i^1\}_{i=1}^N, \{\mathbf{z}_i^2\}_{i=1}^N) + \omega\cdot\ell(\{\mathbf{m}_i^3\}_{i=1}^N, \{\mathbf{m}_i^4\}_{i=1}^N) \tag{21}$$

Notice that, we treat sample pairs that share partial graph identities (*e.g.*, $(\mathbf{m}_j^3, \mathbf{z}_j^1)$ and $(\mathbf{m}_j^3, \mathbf{m}_{\pi(j)}^4)$) as semi-positives and exclude them from negative samples.

**Alignment and Uniformity Loss.** We use dropNode as the intra-graph augmentation and graph interpolation as the cross-graph augmentation. We pre-train GraphCL for 1000 steps before evaluation. The GraphCL setup follows that in Table 10. We split the original pre-training dataset into two subsets. The 80% subset is used for pre-training, and 51200 samples from the other 20% subset are used for loss evaluation.

### E.3 HYPER-PARAMETERS

**Unsupervised Learning.** The hyper-parameters are shown in Table 9. We use the same three-layer GIN from (You et al., 2020). Following Zbontar et al. (2021), the BarlowTwins backbone uses a three-layer MLP projector with hidden dimensions that are four times the input dimension. Following You et al. (2020), we use a learning rate 0.01, batch size 128 and no weight decay. For E-GCL, we conduct grid search for $\alpha$ and $\omega$ in the following ranges $\alpha = \{0.1, 0.4, 1, 2, 4\}$, $\omega = \{0.1, 0.2, 0.3, 0.4, 0.5\}$. We train the GNN for 60 epochs and evaluate the generated embedding using non-linear SVMs every 10 epochs. Following You et al. (2020), we search for the regularization parameter of SVMs in $\{0.001, 0.01, 0.1, 1, 10, 100, 1000\}$. Following (You et al., 2021), we report test accuracy of the epoch selected by validation set.

**Transfer Learning.** The hyper-parameters are shown in Table 10. We use a large batch size 2048 for BarlowTwins and SimSiam to speed up pre-training. The large batch size increases no complexity because Barlowwins and SimSiam use no negative samples. Following Zbontar et al. (2021), the BarlowTwins uses a three-layer MLP projector with hidden dimensions that are four times the input

Table 12: Hyperparameters for reproducing baselines. Parentheses include the original range of hyperparameters if they are different from our reproduction.

(a) Hyperparameters for unsupervised learning baselines.

| | GNN layers | h-dim | Max Epochs | Learning rate | Metric | Epoch selection |
|---|---|---|---|---|---|---|
| InfoGraph | 4 (4,8,12) | 32 | 60 (100) | 1e-3 (1e-2,1e-3,1e-4) | Accuracy | Validation set |
| GraphCL | 3 | 32 | 60 (20) | 1e-2 | Accuracy | Validation set |
| JOAO | 3 | 32 | 60 (40) | 1e-3 | Accuracy | Validation set |
| ADGCL | 5 | 32 | 60 (150) | 1e-3 (1e-2,5e-3,1e-3) | Accuracy | Validation set |
| GraphMAE | 3 (2,3,5) | 32 (32,256,512) | 60 (300) | 1.5e-4 (1.5e-4,5e-4,1e-3) | Accuracy (F1) | Validation set (Last epoch) |
| RGCL | 3 | 32 | 60 (40) | 1e-2 | Accuracy | Validation set |

(b) Hyperparameters for transfer learning baselines.

| | GNN layers | h-dim | Pretrain epochs | Pretrain learning rate | Fine-tune epochs | Epoch selection |
|---|---|---|---|---|---|---|
| Infomax | 5 | 300 | 100 | 1e-3 | 100 | Last epoch (Validation set) |
| ContextPred | 5 | 300 | 100 | 1e-3 | 100 | Last epoch (Validation set) |
| ADGCL | 5 | 300 | 100 (20,50,80,100) | 1e-3 | 100 | Last epoch (Validation set) |
| GraphLOG | 5 | 300 | 1 local + 10 global | 1e-3 | 100 | Last epoch |
| GraphMAE | 5 | 300 | 100 | 1e-3 | 100 | Last epoch |
| RGCL | 5 | 300 | 100 | 1e-3 | 100 | Last epoch |

Table 13: Statistics of unsupervised learning datasets.

| Dataset | Category | #Graphs | #Avg. Node | #Avg. Edges |
|---|---|---|---|---|
| NCI1 | Biochemical Molecules | 4110 | 29.87 | 64.6 |
| PROTEINS | Biochemical Molecules | 1113 | 39.06 | 145.63 |
| DD | Biochemical Molecules | 1178 | 284.32 | 1431.32 |
| MUTAG | Biochemical Molecules | 188 | 17.93 | 39.59 |
| COLLAB | Social Networks | 5000 | 74.49 | 4914.43 |
| RDT-B | Social Networks | 2000 | 429.63 | 995.51 |
| RDT-M | Social Networks | 4999 | 508.52 | 1189.75 |
| IMDB-B | Social Networks | 1000 | 19.77 | 193.06 |

Table 14: Statistics of biochemical graph datasets for transfer learning.

| Datasets | Utilization | #Graphs | #Avg. Node | #Avg. Edges |
|---|---|---|---|---|
| ZINC-2M | Pre-training | 2000000 | 26.62 | 57.72 |
| BBBP | Fine-tuning | 2039 | 24.06 | 51.91 |
| Tox21 | Fine-tuning | 7831 | 18.57 | 38.59 |
| ToxCast | Fine-tuning | 8576 | 18.78 | 38.52 |
| SIDER | Fine-tuning | 1427 | 33.64 | 70.72 |
| ClinTox | Fine-tuning | 1477 | 26.15 | 55.77 |
| MUV | Fine-tuning | 93087 | 24.23 | 52.56 |
| HIV | Fine-tuning | 41127 | 25.51 | 54.94 |
| BACE | Fine-tuning | 1513 | 34.08 | 73.72 |

dimension. The SimSiam (Chen & He, 2021) uses a two-layer MLP projector and a two-layer MLP predictor. Following Tian et al. (2021), we let the predictor network use a learning rate that is ten times that for the GNN encoder and the projector. The SimSiam backbone does not perform well with the default learning rate of $0.001$ and weight decay of $0$. Thus, we search for the SimSiam backbone's learning rate and weight decay values in the following ranges: learning rate $= \{1e-3, 5e-4\}$, weight decay $= \{1e-4, 1e-5, 5e-5\}$. We then fix the learning rate and weight decay and add the equivariance mechanism. For E-GCL, we search for $\alpha$ and $\omega$ from a random subset of the following ranges $\alpha = \{0.1, 0.4, 1, 2, 4\}$, $\omega = \{0.1, 0.2, 0.3, 0.4, 0.5\}$. We do not conduct a grid search because the dataset is large. For fine-tuning, the pre-trained model are re-trained 100 epochs on the transferred dataset. The fine-tuning learning rate is $1e-3$ for GINE and $1e-4$ for GraphTrans. Following (You et al., 2020), we report test performance of the last epoch.

Table 11 shows the configuration details of our used GNNs, in which the GINE is from (Hu et al., 2020) and the GraphTrans is from the Molpcba experiment of (Wu et al., 2021).

Table 15: Complexity of E-GCL and representative GCL baselines. $O(X) = O(2BL(NF^2 + EF))$.

|  | GraphCL | BarlowTwins | SimSiam | RGCL | E-GCL |
|---|---|---|---|---|---|
| GNN encoding | $O(X)$ | $O(X)$ | $O(X)$ | $O(2X)$ | $O(2X)$ |
| Loss function | $O(B^2 F)$ | $O(BF^2)$ | $O(BF)$ | $O(2B^2 F)$ | $O(2BF^2)$ |

Table 16: Complexity of graph augmentations.

| Drop Node | Drop Edge | Subgraph | $\mathcal{G}$-mixup | Graph Interpolation (Ours) |
|---|---|---|---|---|
| $O(N + E)$ | $O(E)$ | $O((1 + kD)N + E)$ | $O(N^3)$ | $O(NF + EF)$ |

**Baseline Hyperparameters.** We have re-evaluated some baselines to present a consistent experimental comparison with E-GCL. In the re-evaluation, we report the test performance selected by validation set for unsupervised learning; we report the last epoch test performance for transfer learning. When reproducing baselines, we change only the evaluation setting and leave other hyperparameters unchanged as much as possible. Table 12 summarizes hyperparameter details. We now describe the process of selecting the baseline hyperparameters. For unsupervised learning, we use the validation set to make decisions when a range of hyperparameters are provided in the original paper. We use the same set of hyperparameters for all datasets. For transfer learning, we fine-tune the same pre-training checkpoint for all downstream datasets for fair comparison.

### E.4 DATASET STATISTICS

Table 13 and Table 14 present the statistics of our used datasets.

## F COMPLEXITY ANALYSIS

The equivariant principle is computationally affordable. In this section, we analyze the complexity of GCL methods in two parts: 1) neural computation, and 2) data augmentation.

**Symbols.** Formally, we define the symbols as follows: $B \in \mathbb{Z}^+$ is the batch size; $N \in \mathbb{Z}^+$ is the number of nodes in a graph; $E \in \mathbb{Z}^+$ is the number of edges in a graph; $L \in \mathbb{Z}^+$ is the number of GNN layers; $k \in (0, 1)$ is the ratio of the cutted subgraph for subgraph augmentation; $D \in \mathbb{Z}^+$ is the maximum degree of nodes in graph; $F \in \mathbb{Z}^+$ is the dimension of features of nodes and edges. For graph interpolation of two graphs, let $N$, $E$, and $D$ refer to the values of the bigger graph.

**Complexity of Neural Computation.** We consider the complexity of GNN encoding and SSL loss function. Here, we analyze the complexity when using the GIN architecture. The E-GCL uses the BarlowTwins framework. Let $O(X) = O(2BL(NF^2 + EF))$ be the complexity of encoding two batches of intra-graph augmentation graphs.

Shown by Table 15, E-GCL's complexity is comparable to RGCL. E-GCL's complexity is at most twice of BarlowTwin. The additional $O(X)$ complexity of GNN encoding comes from encoding the cross-graph augmentation batch, whose size is at most the sum of the two intra-graph augmentations batches. Similarly, E-GCL's complexity of loss function is twice of that of BarlowTwins.

**Complexity of Graph Augmentation.** Table 16 shows the complexity of some popular graph augmentations (You et al., 2020; Han et al., 2022). Note that the complexity of our Graph Interpolation is linear to the graph size times the feature dimension. It is lower than $\mathcal{G}$-mixup and scalable to large graphs with thousands of nodes. Although the complexity of graph interpolation is higher than Drop Node and Drop Edge, it is scalable to large datasets. In practice, a PyTorch dataloader with 4 multiprocessing workers can process graph interpolation of 2048 chemical molecules in batches without putting GPU on wait.

