# OpenReview forum: "Towards Equivariant Graph Contrastive Learning via Cross-Graph Augmentation"
_ICLR.cc/2023/Conference — Submitted to ICLR 2023_

### Official Review · Reviewer_LCfW · 2022-10-23

**Confidence:** 3
**Clarity, Quality, Novelty And Reproducibility:** The proposed method needs more detail…
**Correctness:** 3
**Technical Novelty And Significance:** 3
**Empirical Novelty And Significance:** 2
**Recommendation:** 6

**Strength And Weaknesses:**

Pros:
1. The question studied is important and interesting.
2. Using equivariance can help resolve the sensitive augmentation.

Cons:
1. In Cross-graph augmentation, how do the authors generate labels for instances as
labels are unknown in contrastive learning. The authors claim that they mitigate structural differences between two graphs. However, randomly padding virtual nodes and edges may change the latent distribution of two graphs.
2. In experimental settings, the details of evaluation methods and reported baselines performance are missing. E.g. how do the authors report the baselines performance? Are they reimplementing all baselines or they reporting performances from original papers? Which accuracy is refer to the reported accuracy? The last epoch accuracy or the test accuracy in terms of the best validation accuracy? It is really hard to reproduce the reported performance without these details.
3. As far as I know, the reported results are from different evaluation methods. InfoMax and ContextPred are reporting the test accuracy of the best validation epoch, while GraphLOG is reporting the performance of the last epoch. The authors need to report results under a same experimental settings.
4. In results tables, do the authors use a same backbone for all baselines? As authors mentioned, they are using a BarlowTwins as backbone for unsupervised task and employing a GraphTrans for transfer learning task. Do they also using these backbones for other baselines?

**Summary Of The Paper:**

This paper proposes a cross-graph augmentation method to simulate global semantic shifts. In particular, the paper first analyzes the limitations of invariant graph contrastive learning (I-GCL). Then the authors explore equivariance for cross-graph augmentation to mitigate the limitations of I-GCL. The authors conduct both unsupervised learning and transfer learning experiments the effectiveness of the proposed method.

**Summary Of The Review:**

Overall, this paper takes one the most important research problems of graph learning: graph contrastive. The idea of exploring equivariance cross graph augmentation is interesting. However, I feel this paper is not ready as there are several major concerns regarding the framework design, experiment. Hopefully, the authors can address these concerns and make this paper more solid in the next submission.

##
Score updated after reading the rebuttal.

---

> ### Author Response · Authors · 2022-11-18
> **Response to Reviewer LCfW (Part 3/3)**
>
> > **Comment4: The details of evaluation methods and reported baselines performance are missing.**
> >
> > 1) **Comment4.1: Source of baseline performance.** How do the authors report the baseline performances? Are they reimplementing all baselines or they reporting performances from original papers?
> > 2) **Comment4.2: How to select the reported accuracy?** Which accuracy is refer to the reported accuracy? The last epoch accuracy or the test accuracy in terms of the best validation accuracy?
>
> **Response for Comment4.1:** Thanks for the questions. For the re-evaluation of baselines, we have included the hyperparameter details in Table 12 in Appendix E.3. For the results of our first submission, we have indicated the sources of baseline performances in the caption of Table 8 (original Table 1 in our first submission). For your interest, we present the details as follows:
>
> * **Unsupervised learning:** We borrow the results of No Pre-train, ADGCL and RGCL from [7]. We reproduce the results of GraphMAE using their released source codes. Other baseline performances are borrowed from [8].
>
> * **Transfer learning:** We borrow the results of ADGCL, GraphLOG, and RGCL from [7]. We reproduce the results of GraphMAE using their released source codes. Other baseline performances are borrowed from [8].
>
> **Response for Comment4.2:** Thanks for the comments. We agree that it is important to include the method of selecting accuracy. We have revised Section 4.2 to include the details. For your interest, the details are shown as follows:
>
> * **Unsupervised learning:** Following [7,8,9], we report E-GCL's performances using the test accuracy in terms of the best validation accuracy.
> * **Transfer learning:**  Following [5,8,10], we report E-GCL's performances using the last epoch test performance. This is because of the significant distribution shift between the validation set and the test set in the transfer learning benchmark  [11], making the validation performance and test performance hardly correlated. Thus, E-GCL and other works do not use the validation set to select test performance.
>
>
>
> >  **Comment5: In results tables, do the authors use a same backbone for all baselines?** As authors mentioned, they are using a BarlowTwins as backbone for unsupervised task and employing a GraphTrans for transfer learning task. Do they also using these backbones for other baselines?
>
> Yes, we use the same GNN architecture backbone for all baselines. The details are in Table 12 in Appendix E.3. In specific, BarlowTwins [13] is an SSL loss function. We select BarlowTwins as our invariance loss function, which cannot be adapted to baselines. For unsupervised learning, the baselines use the default three-layer GIN architecture backbone. ADGCL and InfoGraph use different layer numbers due to the recommendation in their original paper. For transfer learning, E-GCL and all baselines use a five-layer GINE from [11]. To fully explore the potentiallity of E-GCL, we also report the performance of E-GCL using the GraphTrans [12] architecture (i.e., E-GCL, GTS) in transfer learning.
>
>
>
> **Reference:**
>
> [1] Demystifying Contrastive Self-Supervised Learning: Invariances, Augmentations and Dataset Biases. In NeurIPS 2020.
>
> [2] Chaos is a ladder: A new theoretical understanding of contrastive learning via augmentation overlap. In ICLR 2022.
>
> [3] A Simple Framework for Contrastive Learning of Visual Representations. In ICML 2020.
>
> [4] A Fine-Grained Analysis On Distribution Shift. ICLR 2022.
>
> [5] Graph Contrastive Learning with Augmentations. NeurIPS 2020.
>
> [6] Connect, Not Collapse: Explaining Contrastive Learning for Unsupervised Domain Adaptation. In ICML 2022.
>
> [7] Let Invariant Rationale Discovery Inspire Graph Contrastive Learning. ICML 2022.
>
> [8] Graph Contrastive Learning Automated. ICML 2021
>
> [9] Adversarial Graph Augmentation to Improve Graph Contrastive Learning. NeurIPS 2021.
>
> [10] Self-supervised Graph-level Representation Learning with Local and Global Structure. ICML 2021.
>
> [11] Strategies For Pre-Training Graph Neural Networks. In ICLR 2020.
>
> [12] Representing Long-Range Context for Graph Neural Networks with Global Attention. In NeurIPS 2021.
>
> [13] Barlow Twins: Self-Supervised Learning via Redundancy Reduction. ICML 2021.

---

> ### Author Response · Authors · 2022-11-18
> **Response to Reviewer LCfW (Part 2/3)**
>
>
> > **Comment2: Randomly padding virtual nodes and edges may change the latent distribution of two graphs.** However, the authors claim that they mitigate structural differences between two graphs.
>
> Good point! We pad virtual nodes to make two graphs have the same size (i.e., mitigate the structural difference). Without this step, graph mixup is impossible due to the irregular sizes of different graphs.
>
> We agree that data augmentations, including graph interpolation and padding nodes, could cause distribution shifts between the augmentation and the original data [1,2,3]. However, the effect of distribution shifts on representation learning is profound and controversial. On one hand, inappropriate distribution shifts (e.g., aggressive augmentations) can change the underlying label and damage representations [1, 2]. On the other hand, distribution shifts can improve the generalization ability when used carefully [3, 4]. We now present our perspectives:
>
> * **Appropriate distribution shifts by data augmentations lead to better performance.** In CV domains, color distortion and mixup applied on images create out-of-distribution augmentations [3]. In graph domains, intra-graph augmentations (e.g., node dropping, subgraph masking) on molecular graphs create new graphs that might not exist in the real world [5]. Despite bringing in distribution shifts, these augmentations improve the generalization ability, because they create overlaps between different distributions or unseen distributions [2,6].
> * **Padding nodes and edges of 0 feature does not change graph semantics.** The padded nodes and edges exist only "virtually" to accommodate the interpolation with another graph. In GNN encoding, these virtual nodes and edges of **0** feature will not be counted and will not change graph embeddings. In Table 5 of Appendix D.1, we show an experiment to demonstrate that the influence to performance when padding more virtual nodes is insignificant. In short, we deliberately mix the largest graphs with the smallest graphs in batch to create size differences. Shown by the Table 5 below, we can see that the performance difference when padding more virtual nodes is only 0.04%.
>
> **Table 5: Unsupervised learning accuracies (%) of E-GCL on the TU datasets. See Appendix D.1 for the full results on 8 datasets.**
>
> |                              | AVG   |
> | ---------------------------- | ----- |
> | E-GCL                        | 76.93 |
> | E-GCL, mixup different sizes | 76.89 |
>
> In future works, we will conduct more investigation on the impact of distribution shifts on graph representation learning.
>
>
>
> >  **Comment3: The authors need to report results under a same experimental settings.** The reported results are from different evaluation methods. InfoMax and ContextPred are reporting the test accuracy of the best validation epoch, while GraphLOG is reporting the performance of the last epoch.
>
> Thanks for the insightful suggestion. We agree that the graph self-supervised learning benchmarks require a re-evaluation that uses a consistent evaluation setting. Thus, we follow your suggestion and conduct a consistent evaluation of baselines for both unsupervised learning and transfer learning. The evaluation setting is the same as E-GCL. For unsupervised learning, we report the test accuracy selected by the validation set for all baselines. For transfer learning, we report the last epoch ROC-AUC for all baselines.
>
> Details of baseline hyperparameters are included in Table 12 in Appendix E.3. Specifically, we reproduce all the baselines except GraphCL and JOAO in transfer learning. This is because the original experimental setting of GraphCL and JOAO is the same as our reproduction, and our computation resource is limited. Thus, we borrow their results from the original papers.
>
> We have updated the Table 1 in our submission with the new consistent baseline results. The original Table 1 from our first submission is moved to AppendixD.4 for your reference. **Due to the presentation limitation of openreview, we cannot present Table 1 here. Please gently find the updated Table 1 in the revised PDF file.**
>
> We want to highlight that our major observations in Section4.2 still hold in the new results. E-GCL remains the state-of-the-art in all GCL methods for unsupervised learning and transfer learning.

---

> > ### Comment · Reviewer_LCfW · 2022-12-03
> > **Node padding**
> >
> > The node padding may not have significant impact on your current tasks but it is not guaranteed in general. In message-passing GNNs, the connections and bias can cause biased learning features. The added virtual node can change the degree of each node, which changes the weights in GCN. And if you use bias in linear transformation, extra biases will be passed through the virtual nodes.

---

> > > ### Author Response · Authors · 2022-12-04
> > > **Response to Node Padding (Part 2/2)**
> > >
> > > **Response for Comment 8.2-Influence on node degree.** Good question! Virtual nodes do not change the degrees of other nodes. The reason is that the degree of a virtual node is 0, because it is not connected to any other nodes by definition.
> > >
> > > For a detailed explanation, we consider the following four cases that involve virtual nodes. Note that, we have $\text{degree}(v) = \lambda *\text{degree}(p) + (1-\lambda) *\text{degree}(q)$, when node $v$ is the interpolation of node $p$ and node $q$. $\lambda$ is the interpolation coefficient.
> > >
> > > * **$v$ is a virtual node.** This cannot happen. A virtual node is included only if it will be interpolated with a real node. Pure virtual nodes are not used for training.
> > > * **$v$ is an interpolation of a real node $p$ and a virtual node $q$.** In this case, $\text{degree}(v)=\lambda \text{degree}(p) + (1-\lambda) \text{degree}(q)=\lambda \text{degree}(p)$. The virtual node $q$ has zero degree, thus it cannot change the degree of $v$. $\text{degree}(v)$ equals to $\lambda \text{degree}(p)$ because we have scaled the local structure of node $p$ for interpolation.
> > > * **$v$ is a real node connected to a virtual node.** This cannot happen because, by definition, virtual nodes are not connected to any other nodes. Also, pure virtual nodes are not used for training.
> > > * **$v$ is a real node connected to a node $u$ that is an interpolation of a real node and a virtual node.** In this case, $A_{vu}=\lambda$ or $1-\lambda$, which is the interpolation coefficient of the real node. By definition, $\text{degree}(v)=A_{vu}+\sum_{p\in N(v)/ \{u\}}A_{vp}$. Thus, $u$ 's contribution to $v$'s degree is properly scaled by the ratio of the real node.
> > >
> > > We conclude that virtual nodes do not change the degree of other nodes. Thus, it does not affect the weights in GCNs.
> > >
> > >
> > >
> > > **Response for Comment 8.3-Extra bias in node update function.** This is a very insightful question on the capacity of Neural Networks. In short, the bias terms can adapt to sparse features, *i.e.*, the interpolation of a real node and a virtual node. Pure virtual nodes are not used for training because they will be interpolated with real nodes. Similar methods in CV have been demonstrated to work successfully when padding 0s for images [1,2]. Note that the node update function in GIN is the MLP function in the following equation:
> > >
> > > $h_u^{(k)} = \text{MLP}^{(k)}\left(\sum_{v\in \mathcal{N}(u)\cup \{u\}} A_{uv}\cdot m_{uv}^{(k)} \right)$
> > >
> > > We want to use an analogy to the [padding in Convolutional Neural Networks (CNNs)](https://en.wikipedia.org/wiki/Convolutional_neural_network#Padding) for answering. In CNNs [1,2], 0-valued pixels are padded on the border of images to let convolution filters slide through the non-zero pixels in the margins. An image patch in the margins is similar to a node that is an interpolation of a real node and a virtual node: their features are sparse and are mixtures of zeros and non-zeros. Despite the sparsity of features, CNNs [1,2] use full convolution filters and full bias terms for these patches in the margins. The successes of CNNs demonstrate that Neural Networks can learn to adapt to sparse inputs that contain 0 entries. Thus, we conclude that the bias terms in the node update function are not **extra** for interpolation nodes.
> > >
> > >
> > >
> > > **Reference:**
> > >
> > > [1] Deep Residual Learning for Image Recognition. In CVPR 2016.
> > >
> > > [2] Learning deep CNN denoiser prior for image restoration. In CVPR 2017.
> > >
> > > [3] Intrusion-Free Graph Mixup. In Arxiv 2022.

---

> > > ### Author Response · Authors · 2022-12-04
> > > **Response to Node Padding (Part 1/2)**
> > >
> > > Thanks for the insightful comment! By our understanding, we can divide your comment into the following three questions.
> > >
> > > > **Comment8:**
> > > >
> > > > 1. **Comment 8.1-Bias in message passing.** When aggregating messages from a virtual node, does the bias term in linear transformation cause biased learning features?
> > > >
> > > > 2. **Comment 8.2-Influence to node degree.** Does the virtual node change the degree of each node? If yes, it will change the weights in GCN.
> > > >
> > > > 3. **Comment 8.3-Extra bias in node update function.** Do extra biases in the node update function pass through the virtual nodes?
> > >
> > > **Response for Comment 8.1-Bias in message passing.** Great catch! It is important to note that the bias term after linear transformation does not cause biased learning features. This is because the message $m_{uv}$ from node $u$ to node $v$ is scaled by the edge weight $A_{uv}$ in the adjacency matrix. The bias term will be scaled smaller when the message comes from a node that is an interpolation of a real node and a virtual node.
> > >
> > > Specifically, let $m_{uv}^{(k)}=Wh_v^{(k-1)}+b$ be an affine transformation (e.g., linear transformation and adding bias) of $h_v^{(k-1)}$. For padded graphs and graph interpolation augmentations, we use the GIN for weighted graphs [3], which is presented by the equation below:
> > >
> > > $h_u^{(k)} = \text{MLP}^{(k)}\left(\sum_{v\in \mathcal{N}(u)\cup \{u\}} A_{uv}\cdot m_{uv}^{(k)} \right) = \text{MLP}^{(k)}\left(\sum_{v\in \mathcal{N}(u)\cup \{u\}} A_{uv}\cdot (Wh_v^{(k-1)}+b) \right)$
> > >
> > > The scaling term $A_{uv}$ guarantees that the aggregated messages will include a proper amount of bias terms. To see this more clearly, we consider four cases for the message $m_{uv}$:
> > >
> > > * **$v$ is a pure virtual node.** This cannot happen. Virtual node is included in our method only if it will be interpolated with a real node. Pure virtual nodes are not used for training.
> > > * **$v$ is an interpolation of a real node and a virtual node.** In this case, $A_{uv}=\lambda$ or $1-\lambda$, which is the graph interpolation coefficient of the real node. Thus, the bias term in $m_{uv}$ will be scaled properly by the ratio of the real node.
> > > * **$v$ is an interpolation of two real nodes.** In this case, $A_{uv}=1$. The bias term safely passes through. This is OK because there is no virtual node involved.
> > > * **$v$ is a real node.** In this case, $A_{uv}=1$. This is exactly the same as the original GIN.
> > >
> > > We conclude that virtual nodes do not cause biased learning features.

---

> ### Author Response · Authors · 2022-11-18
> **Response to Reviewer LCfW (Part 1/3)**
>
> Thanks for your time and insightful suggestions. Following your suggestion, we have re-evaluated the baselines in unsupervised learning and transfer learning under the same setting (Table 1). We have updated our submission to incorporate the new results and your other suggestions. We hope our updated version can resolve you concerns. Looking forward to more discussion with you!
>
>
>
> >  **Comment1: In Cross-graph augmentation, how do the authors generate labels for instances as labels are unknown in contrastive learning?**
>
> Thanks for the insightful question. It is important to note that E-GCL does not use any downstream labels in the pretraining phase. Thus, it is challenging to correctly supervise the cross-graph augmentations. As one of our contributions, we supervise the cross-graph augmentation by interpolating the representations of their original samples. Specifically, we minimize the distance between the two sides of the equation (Equation (7) in paper) below:
> $$
> \phi(\lambda g + (1-\lambda) g') \approx \lambda \phi(g) + (1-\lambda) \phi(g')
> $$
> where $g$ and $g'$ are two graphs, $\lambda\in (0,1)$ is the interpolation ratio, and $\phi$ is a GNN. Intuitively, this method encourages the representation of cross-graph augmentation to equivariantly reflect the semantics of the two input graphs. It encourages the sensitivity for global semantic shifts. For more implementation details, please refer to the Section 3.3 and Section 3.4 of our submission.

---

> > ### Comment · Reviewer_LCfW · 2022-12-03
> > **Labels for generated graph**
> >
> > Thanks for your detailed response.
> >
> > However, my concern is how to ensure the original essential information is maintained. For example, suppose you have two graphs: g1 is toxic and g2 is not toxic in downstream tasks. When you make an interpolation between them, how to know the essential information (toxic or not) in this case? Or how to interpret the interpolated graph things. What kind of information does the interpolated graph contain? The interpolation works well on continuous space. But how does it work on discrete case like graphs is interesting and deserve more attention.

---

> > > ### Author Response · Authors · 2022-12-04
> > > **Response to the Labels for Generated Graphs**
> > >
> > > Thanks for the discussion! By our understanding, we have divided your comment into Comment 6 and Comment 7 below.
> > >
> > > > **Comment 6: How to ensure the original essential information is maintained?** For example, suppose you have two graphs: g1 is toxic and g2 is not toxic in downstream tasks. When you make an interpolation between them, how to know the essential information (toxic or not) in this case? ... What kind of information does the interpolated graph contain?
> > >
> > > Thanks for the profound questions on graph interpolation. We answer the questions from the perspectives of information theory and intuition below:
> > >
> > > **Information theory:** The Intrusion-Freeness Theorem in [1] has proved the recoverability of the two input graphs for graph interpolation. Thus, the graph interpolation should contain information no less than the information of the two original graphs. In other words, the original essential information of the two input graphs is kept in the graph interpolation.
> > >
> > > **Intuition:** Note that, the substructure that decides toxicity (e.g., a cyano group $-C\equiv N$) will be kept still after interpolation, but with smaller weights on nodes and edges. A powerful enough GNN can detect this substructure (Proposition 2 in our submission) and assign smaller confidence on toxicity than that of the original toxic graph g1. In other words, the interpolation should be partially toxic and partially nontoxic. One might argue that the concept of "partially toxic and partially nontoxic" do not exist. However, image interpolations also generate unrealistic images. For example, Figure 1 in [2] is unrealistic for the real world: there is no such thing as "a partially Saint Bernard dog and partially poodle dog".
> > >
> > > To the end, our humble equivariant implementation of graph interpolation is an augmentation to regularize the learning. That is, the augmentation can be less accurate in fidelity sometimes. Other augmentations (e.g., image interpolation and graph node dropout) are not 100% accurate as well. However, the deficiency in fidelity does not undermine their values for training neural networks. The effectiveness of graph interpolation for self-supervised learning has been demonstrated by our experiments in Section 4.
> > >
> > >
> > >
> > > > **Comment 7: How to interpret the interpolated graph things?** The interpolation works well on continuous space. But how does it work on discrete case like graphs is interesting and deserve more attention.
> > >
> > > Indeed, interpreting the graph interpolations is an interesting problem. In this ICLR submission, we stick to the linear interpolation heuristic from previous mixup literatures [1, 2, 3] -- when features are interpolated linearly, their labels should be interpolated linearly as well. For the discrete adjacency matrix, we project it into the continuous space by using weighted graphs. We let the message passing be scaled by the weights of edges. More details of the weighted message passing can be found in the answer of Comment 8.1. In summary, we have adapted the linear interpolation to the discrete graphs, and approximated the semantics of the interpolated graph by the linear interpolation of the semantics of the two original graphs.
> > >
> > > In our case, the linear heuristic works very well. In Table 1, we show that E-GCL achieves state-of-the-art. In Table 2, ablation studies show that E-GCL can be generalized to different GCL frameworks. In our view, linear interpolation is a simple and effective approximation, like the first and second order terms in a Taylor expansion: they are good enough guesses but not 100% accurate.
> > >
> > > For a more rigorous theoretical understanding of the generated graph interpolations, one may need to dive deep into graph theory research. Related topics include graph clique-sum and graph decompositions [4, 5, 6]. However, we respectively argue that it is beyond the scope of this ICLR submission. This submission focuses on graph interpolation's effectiveness for equivariant self-supervised learning.
> > >
> > >
> > >
> > > **Reference:**
> > >
> > > [1] Intrusion-Free Graph Mixup. In arxiv 2022.
> > >
> > > [2] CutMix: Regularization Strategy to Train Strong Classifiers. In ICCV 2019.
> > >
> > > [3] mixup: Beyond Empirical Risk Minimization. In ICLR 2018.
> > >
> > > [4] Clique-sums, tree-decompositions and compactness. Discrete Mathematics. 1990.
> > >
> > > [5] Cycle Decompositions of $K_n$ and $K_{n}-I$. Journal of Combinatorial Theory. 2001.
> > >
> > > [6] Linear-time computation of optimal subgraphs of decomposable graphs. Journal of Algorithms. 1987.

---

> > > ### Author Response · Authors · 2022-12-06
> > > **Follow-up on Labels for generated graph**
> > >
> > > Dear Reviewer LCfW,
> > >
> > > Would you kindly inform us whether our new responses have addressed your concerns? We are looking forward to your feedbacks!
> > >
> > > Authors.

---

> > > ### Author Response · Authors · 2022-12-11
> > > **Window for discussion is closing**
> > >
> > > Dear Reviewer LCfW,
> > >
> > > Thanks again for your time and efforts in reviewing our paper. As the window for discussion will be due in one day (Dec 12), we’d be grateful if you can confirm whether our responses have addressed your concerns. We look forward to more discussions with you if you have any further concerns or questions.
> > >
> > > Best regards,
> > >
> > >
> > > Authors.

---

> ### Author Response · Authors · 2022-11-29
> **Gentle reminder to the reviewer**
>
> Dear Reviewer LCfW,
>
> Thanks again for your efforts in reviewing our paper and the valuable comments. As the window for discussion is closing, we'd be grateful if you can confirm whether our responses have addressed your concerns. We would be glad to have more discussions if you have further questions and comments.
>
> Thanks.
>
> The Authors.

---

### Official Review · Reviewer_f85p · 2022-10-24

**Confidence:** 4
**Correctness:** 4
**Technical Novelty And Significance:** 3
**Empirical Novelty And Significance:** 4
**Recommendation:** 8

**Clarity, Quality, Novelty And Reproducibility:**

This work solves an important problem with a technically sound solution. Conducted experiments are comprehensive to demonstrate the effectiveness of the new approach. The source code has already been released for reproducibility.

**Strength And Weaknesses:**

Pros:
1.	The studied problem with equivariant graph contrastive learning is important and new for GCL domain.
2.	The theoretical discussion of graph interpolation is given.
3.	Component-wise model evaluation is provided to show the model effectiveness.
4.	Hyperparameter sensitivity is conducted in the evaluation section.

Cons:

In the evaluation section, various baselines are compared, such as JOAO, GraphCL, and ADGCL. For the compared various GCL methods, the parameter setting details of those baselines can be described.

Additionally, the computational complexity of the new E-GCL method can be analyzed compared with representative GCL baselines.


**Summary Of The Paper:**

In this work, an equivariant graph contrastive learning method is proposed to perform cross-graph augmentation, which is reasonable and effective for GCL. Experiments are performed both unsupervised learning and transfer learning tasks to show the model effectiveness.

**Summary Of The Review:**

Inspired by the equivariant SSL, this work proposes to enhance graph contrastive learning with cross-graph augmentation under an equivariant SSL framework. Experiments with different tasks show the advantage of the new approach.

---

> ### Author Response · Authors · 2022-11-18
> **Response to Reviewer f85p (Part 2/2)**
>
> > **Comment2: The computational complexity of the new E-GCL method can be analyzed compared with representative GCL baselines.**
>
> We appreciate your insightful suggestion. It is important to see that the equivariant principle is computationally affordable. Thus, we compare the complexity of GCL methods in two parts: 1) neural computation, and 2) data augmentation. The analysis below is included in Appendix F.
>
> **Symbols.** Formally, we define the symbols as follows: $B\in \mathbb{Z}^+$ is the batch size; $N\in \mathbb{Z}^+$ is the number of nodes in a graph; $E\in \mathbb{Z}^+$ is the number of edges in a graph; $L\in \mathbb{Z}^+$ is the number of GNN layers; $k\in (0,1)$ is the ratio of the cutted subgraph for subgraph augmentation; $D\in \mathbb{Z}^+$ is the maximum degree of nodes in graph; $F\in \mathbb{Z}^+$  is the dimension of features of nodes and edges. For graph interpolation of two graphs, let $N$, $E$, and $D$ refer to the values of the bigger graph.
>
>
>
> **Complexity of neural computation.** We consider the complexity of GNN encoding and SSL loss function. Here, we analyze complexity when using the GIN architecture. The E-GCL uses the BarlowTwins framework. Let $O(X)=O(2BL(NF^2 + EF))$ be the complexity of encoding two batches of intra-graph augmentation graphs.
>
> **Table15: Complexity of E-GCL and representative GCL baselines. $O(X)=O(2BL(NF^2 + EF))$.**
>
> |               | GraphCL   | BarlowTwins | SimSiam | RGCL       | E-GCL      |
> | ------------- | --------- | ----------- | ------- | ---------- | ---------- |
> | GNN encoding  | $O(X)$    | $O(X)$      | $O(X)$  | $O(2X)$    | $O(2X)$    |
> | Loss function | $O(B^2F)$ | $O(BF^2)$   | $O(BF)$ | $O(2B^2F)$ | $O(2BF^2)$ |
>
> As shown by the Table above, E-GCL's complexity is comparable to that of RGCL. E-GCL's complexity is at most twice of BarlowTwin's complexity. The additional $O(X)$ complexity of GNN encoding comes from encoding the cross-graph augmentation batch, whose size is at most the sum of the two intra-graph augmentation batches. Similarly, E-GCL's complexity of loss function is twice of that of BarlowTwins.
>
>
>
> **Complexity of graph augmentation.** The Table below shows the complexity of some popular graph augmentations [1,2]. Note that the complexity of our graph interpolation is linear to the graph size times the feature dimension. It is lower than that of $\mathcal{G}$-mixup and scalable to large graphs with thousands of nodes. Although the complexity of graph interpolation is higher than Drop Node and Drop Edge, it is scalable to large datasets. In practice, a PyTorch dataloader with 4 multiprocessing workers can process graph interpolation of 2048 chemical molecules in batches without putting GPU on wait.
>
> **Table16: Complexity of graph augmentations.**
>
> | Drop Node [1] | Drop Edge [1] | Subgraph [1]     | $\mathcal{G}$-mixup [2] | Graph Interpolation (Ours) |
> | ------------- | ------------- | ---------------- | ----------------------- | -------------------------- |
> | $O(N+E)$      | $O(E)$        | $O((1+kD)N + E)$ | $O(N^3)$                | $O(NF + EF)$               |
>
>
>
> **Reference:**
>
> [1] Graph Contrastive Learning with Augmentations. In NeurIPS 2020.
>
> [2] G-Mixup: Graph Data Augmentation for Graph Classification. In ICML 2022.

---

> ### Author Response · Authors · 2022-11-18
> **Response to Reviewer f85p (Part 1/2)**
>
> Thanks so much for your time and positive feedback! Your suggestions have been carefully incorporated into our revision. To address your concerns, we present point-to-point responses as follows.
>
> > **Comment1: The parameter setting details of baselines can be described.** In the evaluation section, various baselines are compared, such as JOAO, GraphCL, and ADGCL.
>
> Thanks for the valuable suggestion. We agree that it is important to report the parameter settings of baseline methods. In the caption of Table 8 (originally Table 1), we have reported the source of the baseline performances in our first submission. Following the suggestion of Reviewer LCfW, we have re-evaluated some baselines for consistent comparison. The parameter settings of the re-evaluation is included in Table 12 in Appendix E.3. For your information, we present Table 12 below. We do not re-evaluate GraphCL and JOAO in transfer learning because they already use the same hyperparameters as our re-evaluation. Thus, we borrow their performances from the original paper directly.
>
> **Table 12 (a): Hyperparameters for reproducing baselines in unsupervised learning.  Parentheses include the original range of hyperparameters if they are different from our reproduction.**
>
> |           | GNN layers | h-dim           | Max epochs | Learning  rate            | Metric        | Epoch selection             |
> | --------- | ---------- | --------------- | ---------- | ------------------------- | ------------- | --------------------------- |
> | InfoGraph | 4 (4,8,12) | 32              | 60 (100)   | 1e-3 (1e-2,1e-3,1e-4)     | Accuracy      | Validation set              |
> | GraphCL   | 3          | 32              | 60 (20)    | 1e-2                      | Accuracy      | Validation set              |
> | JOAO      | 3          | 32              | 60 (40)    | 1e-3                      | Accuracy      | Validation set              |
> | ADGCL     | 5          | 32              | 60 (150)   | 1e-3 (1e-2,5e-3,1e-3)     | Accuracy      | Validation set              |
> | GraphMAE  | 3 (2,3,5)  | 32 (32,256,512) | 60 (300)   | 1.5e-4 (1.5e-4,5e-4,1e-3) | Accuracy (F1) | Validation set (Last epoch) |
> | RGCL      | 3          | 32              | 60 (40)    | 1e-2                      | Accuracy      | Validation set              |
>
> **Table 12 (b): Hyperparameters for reproducing baselines in transfer learning.  Parentheses include the original range of hyperparameters if they are different from our reproduction.**
>
> |             | GNN layers | h-dim | Pretrain epochs     | Pretrain learning rate | Fine-tune epochs | Epoch selection             |
> | ----------- | ---------- | ----- | ------------------- | ---------------------- | ---------------- | --------------------------- |
> | Infomax     | 5          | 300   | 100                 | 1e-3                   | 100              | Last epoch (Validation set) |
> | ContextPred | 5          | 300   | 100                 | 1e-3                   | 100              | Last epoch (Validation set) |
> | ADGCL       | 5          | 300   | 100 (20,50,80,100)  | 1e-3                   | 100              | Last epoch (Validation set) |
> | GraphLOG    | 5          | 300   | 1 local + 10 global | 1e-3                   | 100              | Last epoch                  |
> | GraphMAE    | 5          | 300   | 100                 | 1e-3                   | 100              | Last epoch                  |
> | RGCL        | 5          | 300   | 100                 | 1e-3                   | 100              | Last epoch                  |

---

> ### Author Response · Authors · 2022-11-30
> **Gentle reminder to the reviewer**
>
> Dear Reviewer f85p,
>
> Thanks so much for your efforts in reviewing our paper and the valuable comments. We really appreciate your positive evaluation of both the novelty and significance of our work. As the window for discussion is closing, we'd be grateful if you can confirm whether our responses have addressed your concerns on parameter settings and complexity analysis. We would be glad to have more discussions if you have further questions and comments.
>
> Thanks.
>
> The Authors.

---

### Official Review · Reviewer_itQe · 2022-10-24

**Confidence:** 4
**Correctness:** 3
**Technical Novelty And Significance:** 3
**Empirical Novelty And Significance:** 3
**Recommendation:** 6

**Clarity, Quality, Novelty And Reproducibility:**

This paper is well written with good novelty. And the code is available in the appendix.

**Strength And Weaknesses:**

Strength 1: The contribution of this paper is clear; this paper is well written.
Strength 2: This paper deals with a non-trivial and challenging issue which are neglected by current research.
Strength 3: The experiment was performed on several datasets. The results show superior performance over existing competitive approaches.

Weakness 1: In the introduction part, the authors point out that ‘Nonetheless, it is hard, without domain knowledge (Dangovski et al., 2022; Chuang et al., 2022) or extensive testing (Dangovski et al., 2022), to tell apart sensitive and insensitive augmentations.’  Indeed, it is an unresolved challenge to tell if a data augmentation will cause global semantic shifts. Learning the semantic shifted view can lead to sub-optimal performance. However, how the equivariant self-supervised learning can mitigate this issue? In the proposed framework, the intra-graph augmentation is kept. Hence, the semantic shifted view can still damage the model.

Weakness 2: In Section 3.2, the author applied the ‘Group Averaging’ to ‘make the interpolations insensitive to relative permutations’. However, in the Section 3.1, the author claims that ‘Equivariance requires that…, any change applied to the graph should be faithfully reflected by the change of representation.’ This is not consistent. Can the author give more discussions on why the model should be insensitive to relative permutations during cross-graph augmentation? And why apply the ‘Group Averaging’ can help improve performance? There is no further experiment or discussion in Section 4 to back up this design.

Weakness 3: In Section 4.3, the authors demonstrate the alignment and uniformity losses of I-GCL and E-GCL. It can be observed that the "E-GCL with cross-aug" setting has better alignment and uniformity losses. Can the author give some discussion on this finding? Does this imply if getting rid of the intra-aug, the model may have a better downstream task performance? If possible, the author should conduct ablation study on the "E-GCL with cross-aug".

**Summary Of The Paper:**

This paper first points out that aggressive augmentations may worsen representation and lead to sub-optimal performance. To mitigate this issue, they propose a novel contrastive learning framework (E-GCL) that employs the equivariance principle to implement cross-graph discrimination. To this end, the authors propose insensitive transformations P and sensitive transformations H for the intra-graph augmentation and cross-graph augmentation, respectively. The author conducted detailed experiments to prove their framework’s effectiveness.

**Summary Of The Review:**

This paper is well-written and deals with a non-trivial challenge. The experiments prove the model’s effectiveness. However, theoretical justification for the proposed design and discussion on some experiments are missing. The authors should address these issues. Overall, this paper is above the acceptance borderline if the authors can address all the concerns mentioned above.

---

> ### Author Response · Authors · 2022-11-18
> **Response to Reviewer itQe (Part 2/3)**
>
> > **Comment2: Why the model should be insensitive to relative permutations during cross-graph augmentation?** Why ‘Group Averaging’ can help improve performance? There is no further experiment to back up this design.
>
> Thanks for the valuable comments and suggestions. Note that relative permutations between graphs do not encode useful information about the semantics of the two input graphs. Thus, we require insensitivity to relative permutations to avoid including spurious information in graph representation.
>
> Unlike images and texts, the basic elements in graphs (i.e., nodes) are unordered. Thus, node permutations (orderings) encode nothing useful about graph semantics. This is supported by the permutation invariant design of GNNs [2, 3]. To achieve insensitivity to relative permutations, we randomly permute one of the two graphs before interpolation. Further, in Proposition 1, we prove that this random permutation strategy approximates group averaging, which is strictly invariant to relative permutation.
>
> Following your suggestion, we present experiments to back up this design. We want to show that:
>
> * using random permutation improves performance;
> * using more samples to approximate group averaging improves performance;
> * the improvement of using more samples is only marginal and comes at the cost of increased complexity.
>
>
>
> **Table 6: Unsupervised learning performance in TU datasets. We ablate E-GCL using different numbers of samples to approximate group averaging. We report the average and max of mean accuracies for E-GCL with different $\omega=\{0.1, 0.2, 0.3, 0.4, 0.5\}$. We set $\alpha=0.1$.**
>
> |                   | No rand. perm. | 1-sample | 3-sample | 10-sample |
> | ----------------- | -------------- | -------- | -------- | --------- |
> | Average           | 76.65          | 76.69    | 76.73    | **76.80** |
> | Max               | 76.85          | 76.92    | 76.95    | **76.97** |
> | Forward time (ms) | 10.5±2.4       | 10.7±2.1 | 15.1±3.1 | 29.4±8.0  |
>
> Table 6 above shows that using random permutation consistently outperforms not using random permutation (No rand. perm.). Specifically, the 1-sample estimator performs better and adds no computational cost compared to the No rand. perm.. Further, it shows that using more samples to estimate the group averaging improves performance: the 10-sample estimator gives the best performance. However, the 10-sample estimator's improvement is only marginal ($0.05\% \sim 0.11\%$) compared to the 1-sample estimator, but leads to almost three times increase in time complexity. Thus, we recommend the 1-sample estimator in implementation.
>
> This Table and the detailed experimental settings are included in Appendix D.2.

---

> ### Author Response · Authors · 2022-11-18
> **Response to Reviewer itQe (Part 1/3)**
>
> We gratefully thank you for the positive feedback and constructive comments. To address your concerns, we provide the point-to-point responses. We have carefully revised our paper by taking into account all your suggestions. Looking forward to more discussions with you.
>
> > **Comment1: How the equivariant self-supervised learning can mitigate the issue of aggressive intra-graph augmentation?** In E-GCL, the intra-graph augmentation is kept. Hence, the semantic shifted view can still damage the model.
>
> Thanks for the insightful question targeting at the motivation. To answer the question, we provide intuitions and experiments below.
>
> * **Intuitions:** Intra-graph augmentations are problematic that they sometimes harmfully enforce insensitivity to semantically shifted graphs (i.e., aggressive augmentations). To patch the problem, cross-graph augmentations always enforce sensitivity to semantically shifted graphs that are generated by graph interpolation. Consequently, the equivariance to cross-graph augmentations diminish the harmful invariance of aggressive intra-graph augmentations that change global semantics, leading to better performance.
>
>   We have revised the last paragraph of introduction and Appendix D.3 for clarity.
>
> * **Experiments:** To demonstrate our claim, we have presented experiments in Table 7 at Appendix D.3. Specifically, we compare the Average Confusion Ratio (**ACR**) [1] for GraphCL without and with the equivariance principle. ACR is a metric to measure the ratio that the anchor graph's nearest neighbors are the views of different graphs, rather than the other views of the same anchor. Higher ACR indicates that the graph representations are less powerful to distinguish different graphs, thus reflecting worse negative influences of aggressive augmentations. We copy-paste Table 7 as follows:
>
> **Table 7: ACR with the GraphCL SSL framework. Lower is better.**
>
> |               | Before pre-training | After pre-training |
> | ------------- | ------------------- | ------------------ |
> | GraphCL       | 0.983               | 0.463              |
> | +Equivariance | 0.983               | **0.423**          |
>
> We can observe that applying Equivariance improves the ACR for the GraphCL framework. This result demonstrates that E-GCL's equivariance principle mitigates the negative influences of aggressive augmentations, thus leading to better graph discrimination performances, as compared to I-GCL.

---

> ### Author Response · Authors · 2022-11-29
> **Gentle reminder to the reviewer**
>
> Dear Reviewer itQe,
>
> Thanks again for your efforts in reviewing our paper and the valuable comments. As the window for discussion is closing, we'd be grateful if you can confirm whether our responses have addressed your concerns. We would be glad to have more discussions if you have further questions and comments.
>
> Thanks.
>
> The Authors.

---

### Official Review · Reviewer_sATH · 2022-10-27

**Confidence:** 4
**Clarity, Quality, Novelty And Reproducibility:** The writing is clear, but the novelty…
**Correctness:** 3
**Technical Novelty And Significance:** 2
**Empirical Novelty And Significance:** Not applicable
**Recommendation:** 3

**Strength And Weaknesses:**

Strength:
1. Extensive results on multiple datasets.
Weakness:
1. Novelty: Though the authors make a great effort in explaining the concept of equivalence, the framework itself boils down to a combination of Mixup and contrastive learning. This combination is not new. For instance, at a high level, [1] and [2] both adopt the combination.
2. The experimental results are not very convincing. In table 1(a), on the datasets that the proposed method performs the best, the increase is usually below 1% , not to mention that on the remaining 3 datasets, the proposed method underperforms. Similarly, in table 1(b), the proposed method only wins 4/8 datasets and by a small margin.


Literature:
[1]i-Mix: A Domain-Agnostic Strategy for Contrastive Representation Learning
[2]Improving Contrastive Learning by Visualizing Feature Transformation

**Summary Of The Paper:**

The authors propose an equivariant graph contrastive learning framework that adopts two principles: invariance to intra-graph
augmentations and equivariance to cross-graph augmentations. They use Mixup as the method for cross-graph augmentation and argue that the cross-graph augmentation captures global semantic shifts and yield better performance.

**Summary Of The Review:**

In general, I am not a fan of this paper mainly for the following two reasons: (1) Combining Mixup and contrastive learning is not new as several papers have adopted a similar idea though in the CV domain. (2) The experimental results are not exciting.

---

> ### Author Response · Authors · 2022-11-18
> **Response to Reviewer sATH (Part 2/2)**
>
>
> > **Comment2: The experimental results are not exciting.** In Table 1(a), on the datasets that the proposed method performs the best, the increase is usually below 1% , not to mention that on the remaining 3 datasets, the proposed method underperforms. Similarly, in table 1(b), the proposed method only wins 4/8 datasets and by a small margin.
>
> Thanks for the comments. To our knowledge, E-GCL is the state-of-the-art model on both unsupervised learning and transfer learning settings. To support this claim, we quote other reviewers:
>
> * Reviewer itQE: "The results show **superior** performance over existing competitive approaches"
>
> * Reviewer f85p: "Experiments with different tasks show the **advantage** of the new approach"; "Component-wise model evaluation is provided to show the model **effectiveness**".
>
>
>
> **Table: The number of datasets that the method outperforms every baseline. Numbers are obtained by comparing results in original papers. Bold indicates the best performance.**
>
> | Method   | Unsupervised learning (8 datasets) | Transfer learning (8 datasets) | Total  |
> | -------- | ---------------------------------- | ------------------------------ | ------ |
> | GraphCL  | 4                                  | 3                              | 7      |
> | R-GCL    | 5                                  | 4                              | 9      |
> | JOAO     | 2                                  | 2                              | 4      |
> | GraphLOG | -                                | **6**                          | 6      |
> | E-GCL    | **6**                              | 4                              | **10** |
>
> The table above summarizes the number of improved datasets for representative graph SSL methods. Note that, we have slightly improved the unsupervised learning experiments. Now, E-GCL outperforms baselines in 6 out of 8 datasets in unsupervised learning. We can observe that:
>
> * The experimental improvement of E-GCL is advantageous compared to that of previous works.
> * Previous works do not improve consistently on every dataset of benchmarks.
>
> The problem of inconsistent improvements is caused by the **diversity** of graph datasets. This diversity makes it very challenging to improve consistently on every dataset of the benchmark. In computer vision, SSL methods are evaluated on datasets from a single source without considering subcategories (*e.g.*, ImageNet and COCO). Unlike computer vision, the datasets of graph SSL come from multiple sources of different areas, including social networks, protein-interaction networks, and chemical molecules (Table 1 (a)). Each area can be further divided into domains (Table 1(b)). It is challenging to find a prior that consistently improves on every domain. In this case, the improvement on average performance should be highlighted. Thus, we respectfully argue that E-GCL shows exiciting improvement by establishing state-of-the-art average performance.
>
>
>
> **References:**
>
> [1] i-Mix: A Domain-Agnostic Strategy for Contrastive Representation Learning. ICLR 2021
>
> [2] Improving Contrastive Learning by Visualizing Feature Transformation. ICCV 2021
>
> [3] Hard Negative Mixing for Contrastive Learning. NeurIPS 2020.
>
> [4] IfMixup: Towards Intrusion-Free Graph Mixup for Graph Classification. arXiv 2022.

---

> ### Author Response · Authors · 2022-11-18
> **Response to Reviewer sATH (Part 1/2)**
>
> We appreciate your valuable comments. To address your concerns, we provide point-to-point responses to justify the novelty and the empirical significance of our method. We look forward to having more discussion with you.
>
> > **Comment 1: Combining Mixup and contrastive learning is not new as several papers [1,2] have adopted a similar idea though in the CV domain.** Though the authors make a great effort in explaining the concept of equivalence, the framework itself boils down to a combination of Mixup and contrastive learning.
>
> We agree that mixup and contrastive learning are the major components of our E-GCL framework. However, we respectively argue that E-GCL is novel and has significant differences from prior studies [1,2,3] w.r.t. three aspects: design principles, mixup implementation, and contrastive learning. We summarize the differences in the Table below.
>
> |                             | E-GCL                                 | i-mix [1]                          | [2]                   | MoCHi[3]              |
> | --------------------------- | ------------------------------------- | ---------------------------------- | --------------------- | --------------------- |
> | **Design Principle**        |                                       |                                    |                       |                       |
> | Support                     | Group theory                          | Heuristics                         | Heuristics            | Heuristics            |
> | Intra-instance invariance   | $\checkmark$                          | -                                  | -                     | -                     |
> | Cross-instance equivariance | $\checkmark$                          | -                                  | -                     | -                     |
> | **Mixup Implementation**    |                                       |                                    |                       |                       |
> | Data                        | Graph                                 | Image                              | Image                 | Image                 |
> | Scope                       | Raw feature, hidden representation    | Raw feature, hidden representation | Hidden representation | Hidden representation |
> | Cross-instance              | $\checkmark$                          | $\checkmark$                       | -                     | $\checkmark$          |
> | Group averaging             | $\checkmark$                          | -                                  | -                     | -                     |
> | **Contrastive  Learning**   |                                       |                                    |                       |                       |
> | Branch                      | Invariant branch + Equivariant branch | Single branch                      | Single branch         | Single branch         |
> | Paradigm                    | Cooperative game between two branches | -                                  | -                     | -                     |
> | Effect                      | Mitigate aggressive augmentation      | -                                  | -                     | -                     |
>
> * **Design Principle.** Introducing group theory is important for understanding the complementary property between invariance and equivariance. Thus, it inspires the cooperative game between an invariant branch for intra-graph augmentation and an equivariant branch for cross-graph augmentation. By contrast, other methods are motivated by heuristics and use a single branch for learning.
>
> * **Mixup Implementation.** E-GCL is the only method that is tailored for graphs in mixup implementation. It improves IfMixup [4] and employs group averaging to mitigate the difficulties in graph interpolation.
> * **Contrastive Learning.** E-GCL targets at the aggressive intra-graph augmentation problem in contrastive learning. This problem is not discussed in prior works [1,2,3].
>
> In summary, we argue that E-GCL is a novel method and significantly different from [1,2,3] due to our inspiration from group theory and the tailoring for graph data.

---

> ### Author Response · Authors · 2022-11-29
> **Gentle reminder to the reviewer**
>
> Dear Reviewer sATH,
>
> Thanks again for your efforts in reviewing our paper and the valuable comments. As the window for discussion is closing, we'd be grateful if you can confirm whether our responses have addressed your concerns. We would be glad to have more discussions if you have further questions and comments.
>
> Thanks.
>
> The Authors.

---

> ### Author Response · Authors · 2022-12-11
> **Window for discussion is closing**
>
> Dear Reviewer sATH,
>
> Thanks again for your time and efforts in reviewing our paper. As the window for discussion is closing, we’d be grateful if you can confirm whether our response has addressed your concerns.
>
> If you have further questions after reading the responses and the revised paper, it would be great to let us know. We are happy to address them.
>
> Best regards,
>
> Authors.

---

### Author Response · Authors · 2022-11-18
**General Response and Paper Revision**

We appreciate all the reviewers' efforts for reviewing this submission. Our submission has received diverse ratings, including one accept (8), one borderline accept (6), and two rejects (3). We would like to thank all the reviewers for providing insightful comments and valuable suggestions. We have incorporated the suggestions in the updated submission. The major updates are summarized as follows:

* **New Baseline Results.** Following the suggestion of Reviewer LCfW, we have re-evaluated the baseline performances using a consistent evaluation setting. We have updated Table 1 to include the new results. The original Table 1 from our first submission is moved as Table 8 in Appendix D.4 for your reference. Specifically, the re-evaluation of baselines reports the test accuracy selected by the validation set in unsupervised learning, and reports the last epoch performance in transfer learning. The hyperparameter details of reproducing baselines is summarized in Table 12 in Appdendix E.3.

  Note that, our observations and conclusions in Section 4 remain the same with the new results, which again demonstrate the effectiveness of E-GCL.

* **Improved Unsupervised Learning Results.**  After using learning rate 1e-3, E-GCL's unsupervised learning performances have been slightly improved. Now, E-GCL achieves 76.93% mean accuracy on the TU datasets. E-GCL achieves the best mean accuracy among GCL methods and the best performance in six out of eight datasets for unsupervised learning.

  We have updated Table 1a, Table 2a, Figure 3, Table 3, and Table 5 to include the new results.

* **Ablation Study on Group Averaging.** Following the suggestion of Reviewer itQe, we present experiments on the effect of group averaging in Appendix D.2. The new results demonstrate the effectiveness of using group averaging.

* **Complexity Analysis.** Following the suggestion of Reviewer f85p, we compare the complexity of E-GCL and representative baselines in Appendix F. We show that E-GCL's complexity is comparable to baselines.

* **[Reviewer itQe] Clarification on Equivariance.** The frist paragraph of Section3.1 is revised to clarify that E-GCL encourages equivariance to a **pre-defined** set of transformations, rather than **arbitrary** transformations.

* **[Reviewer itQe] Clarification on Group Averaging.** The group averaging paragraph of Section 3.2 is revised to clarify the motivation.

We have highlighted the updates in the revision. We hope our response and revision can resolve the confusions and lift the concerns. We look forward to your reply!

---

### Author Response · Authors · 2022-12-01
**Ask for your help**

Dear Area Chairs,

We want to draw your attention to our ICLR submission. After carefully considering the reviewers' comments, the major concerns of our submission, in our opinion, are related to the inclusion of more experiments. Thus, we add a large number of experiments in response to the reviewer's comments. We also provide point-to-point answers to address all the concerns of reviewers. Despite the substantial time and effort that we spend on answering the reviewers' questions, there is sadly no discussion for our submission.

The author-reviewer discussion deadline is approaching. We sincerely hope that our efforts and improvements will be taken into consideration. We would appreciate it if you could help us to remind the reviewers.

Thanks for your attention and participation.

Authors

---

### Decision · Program_Chairs · 2023-01-20

**Decision:**

Reject

**Justification For Why Not Higher Score:**

The proposed mixup generating graphs outside of the support with unclear labels is not yet theoretically sound as it should be and needs a more solid theoretical foundation before publication.

**Justification For Why Not Lower Score:**

The work has some interesting ideas on contrastive learning for graphs, and how to deal with permutation invariance and equivariance for these methods.

**Metareview: Summary, Strengths And Weaknesses:**

This work proposes equivariant graph contrastive learning (a contrastive learning framework (E-GCL) ) that adopts two principles: invariance to intra-graph augmentations and equivariance to cross-graph augmentations. The work interestingly shows that aggressive augmentations may worsen representation and lead to sub-optimal performance. The work uses a mixup procedure as the method for cross-graph augmentation and argue that the cross-graph augmentation captures global semantic shifts and yield better performance. Experiments were performed in both unsupervised learning and transfer learning tasks to show the model effectiveness.

One of the reviewers was concerned about the mixup procedure (other reviewers have raised the same concern). Why should the graph mixup be a linear interpolation heuristic? Why would the mixup not account for graph labels? The mixup is also an out-of-distribution augmentation, which is not proven to be safe in most applications. Why is it safe in the applications proposed by the paper? Graph mixup is an interesting topic but the interpolation technique is itself is too naive to deal with the complexity of graphs (interpolation is rarely the answer to finding intermediate points in the support of graph distributions). The authors justify their choices citing other works, but after reading these works, I see the same issue with these other papers.

The were questions about the key differences to prior works [1,2,3] being straightforward (e.g., intra-graph augmentation). On the positive side, the use of permutation averaging for insensitivity to permutation in order to deal with graph interpolation is interesting.

Overall, reviewers raised some questions that were addressed in the rebuttal. However, the challenges with the mixup generating graphs outside of the support with unclear labels does not look theoretically sound (regardless of what authors have argued in the rebuttal, since it needs a proof, not just justifications) and needs to be addressed before publication.